# *Navigating the Digital World as Humans Do:* UNIVERSAL VISUAL GROUNDING FOR GUI AGENTS

**Boyu Gou**[1]  **Ruohan Wang**[1]  **Boyuan Zheng**[1]  **Yanan Xie**[2]  **Cheng Chang**[2]  **Yiheng Shu**[1]
**Huan Sun**[1]  **Yu Su**[1]
[1]The Ohio State University  [2]Orby AI
{gou.43, sun.397, su.809}@osu.edu, yanan@orby.ai
https://osu-nlp-group.github.io/UGround/

## ABSTRACT

Multimodal large language models (MLLMs) are transforming the capabilities of graphical user interface (GUI) agents, facilitating their transition from controlled simulations to complex, real-world applications across various platforms. However, the effectiveness of these agents hinges on the robustness of their grounding capability. Current GUI agents predominantly utilize text-based representations such as HTML or accessibility trees, which, despite their utility, often introduce noise, incompleteness, and increased computational overhead. In this paper, we advocate a human-like embodiment for GUI agents that perceive the environment entirely visually and directly perform pixel-level operations on the GUI. The key is visual grounding models that can accurately map diverse referring expressions of GUI elements to their coordinates on the GUI across different platforms. We show that a simple recipe, which includes web-based synthetic data and slight adaptation of the LLaVA architecture, is surprisingly effective for training such visual grounding models. We collect the largest dataset for GUI visual grounding so far, containing 10M GUI elements and their referring expressions over 1.3M screenshots, and use it to train UGround, a strong universal visual grounding model for GUI agents. Empirical results on six benchmarks spanning three categories (grounding, offline agent, and online agent) show that 1) UGround substantially outperforms existing visual grounding models for GUI agents, by up to 20% absolute, and 2) agents with UGround outperform state-of-the-art agents, despite the fact that existing agents use additional text-based input while ours only uses visual perception. These results provide strong support for the feasibility and promise of GUI agents that navigate the digital world as humans do.

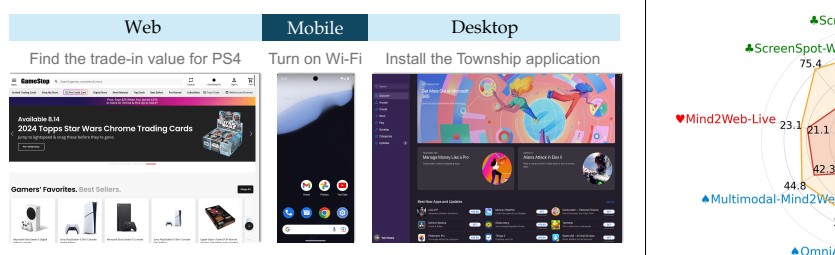

Figure 1: Examples of agent tasks across platforms and performance on **GUI grounding** (♣: ScreenSpot), **offline agent** (♠: Multimodal-Mind2Web, AndroidControl, and OmniACT), and **online agent benchmarks** (♥: Mind2Web-Live and AndroidWorld) when using GPT-4 as the planner.

# 1 INTRODUCTION

GUI (graphical user interface) agents, which are autonomous agents acting in the digital world via operating on GUIs, have been rapidly co-evolving with large language models (LLMs). On the

one hand, the general multimedia understanding and generation capabilities of (multimodal) LLMs empower GUI agents to generalize beyond simple simulated settings (Shi et al., 2017; Humphreys et al., 2022) to diverse and complex real-world environments, including the web (Deng et al., 2023; Zhou et al., 2024; Yao et al., 2022), desktop (Xie et al., 2024; Wu et al., 2024) and mobile operating systems (Rawles et al., 2023; Yan et al., 2023; Rawles et al., 2024). On the other hand, GUI agents have become an important testbed for LLMs, providing both the necessary breadth and depth for driving continued development as well as a pathway to many commercially viable automation applications.

Most humans perceive the digital world visually and act via keyboards, mice, or touchscreens. In principle, the embodiment of a GUI agent should already be *complete* if it can 1) visually perceive the GUI renderings, and 2) have effectors equivalent to a keyboard for typing and equivalent to a mouse or touchscreen for pixel-level operations like clicking and hovering.[1] However, current GUI agents assume more than that. For perception, most current agents rely on reading the underlying text-based representations such as HTML or accessibility (a11y) trees (Deng et al., 2023; Gur et al., 2024; Zhou et al., 2024).[2] Only with the recent advances in multimodal LLMs (MLLMs) does visual perception become broadly viable, but text-based representations are still used jointly (Zheng et al., 2024; Koh et al., 2024; Zhang et al., 2024a). For effectors, most current agents act via selecting from a list of options, e.g., HTML elements (Deng et al., 2023; Zheng et al., 2024) or labeled bounding boxes (He et al., 2024; Zhang et al., 2024a), instead of pixel-level operations directly on the GUI. Obtaining those options in turn often requires access to text-based representations and/or separate models for detecting objects and text (Wang et al., 2024a; Kapoor et al., 2024).

However, there is no free lunch, and those additional requirements come with their limitations. On the one hand, *text-based representations are noisy and incomplete*. Full HTML documents contain a considerable amount of irrelevant information. A11y trees are more compact and mainly contain semantic information, but similar to other semantic annotations that rely on voluntary participation, they widely suffer from incomplete and incorrect annotations.[3] In contrast, visual renderings, by design, are information-complete and only contain information relevant to users. On the other hand, *the additional input increases latency and inference costs*. Zheng et al. (2024) found that HTML can consume up to 10 times more tokens to encode than the corresponding visual. Meanwhile, obtaining an a11y tree can be time-consuming in itself, especially in desktop or mobile environments. The added latency and cost at every step are further compounded in the long-horizon agent tasks, compromising user experience and practicality.

In this work, we are interested in *how far GUI agents with a human-like embodiment, i.e., only visual observation of environments and pixel-level operations, can go*. There have been a few attempts (Shaw et al., 2023; Hong et al., 2024; Cheng et al., 2024), but they are rarely adopted in state-of-the-art solutions. We find that a major bottleneck is *grounding*, i.e., mapping textual plans generated by an (M)LLM to the precise locations on the GUI. There are three desiderata for a GUI agent grounding model: 1) *High accuracy*. A single grounding error can get an agent stuck and fail the whole task. 2) *Strong generalization*. It should work on different GUIs: desktop (Windows, Linux, macOS), mobile (Android, iOS), different websites, etc. 3) *Flexibility*. It should plug and play in different MLLMs instead of being tightly coupled with a certain model. Existing visual grounding methods for GUI agents (Shaw et al., 2023; Hong et al., 2024; Cheng et al., 2024) fail to meet these desiderata, hindering the advances towards GUI agents with human-like embodiment.

The main contributions of this work are three-fold:

1. We make careful arguments and a strong case for GUI agents with human-like embodiment that perceive the digital world entirely visually and take pixel-level operations on GUIs, and propose a generic framework, **SeeAct-V**, for building such agents by adapting from the popular SeeAct framework (Zheng et al., 2024).

---

[1]Except for auditory perception, which is beyond the scope of this study.

[2]The a11y tree is a compact yet informative representation intended for assistive technologies to facilitate people with disabilities, e.g., visual impairment.

[3]A 2024 survey over the top one million websites found that 95.9% of the home pages had accessibility conformance errors such as missing alternative text for images or missing form input labels, with an average of 56.8 errors per page (WebAIM, 2024).

2. We show that a simple recipe, which includes web-based synthetic data and slight adaptation of the LLaVA architecture (Liu et al., 2024c), is surprisingly effective for GUI visual grounding. Using this recipe, we construct and release the **largest GUI visual grounding dataset** to date, covering 10M GUI elements and their referring expressions over 1.3M GUI screenshots. We also train and release a universal visual grounding model, **UGround**, on the dataset.

3. We conduct the most comprehensive evaluation for GUI agents to date, covering six benchmarks spanning three categories (Figure 1): **grounding** (desktop, mobile, and web), **offline agent evaluation** (desktop, mobile, and web), and **online agent evaluation** (mobile and web). The results demonstrate: 1) UGround substantially outperforms existing visual grounding models for GUI agents across the board, by up to 20% absolute. 2) SeeAct-V agents with UGround can achieve at least comparable and often much better performance than state-of-the-art agents that use additional text-based input. These results provide strong support for the feasibility and promises of GUI agents that navigate the digital world as humans do.

## 2 METHOD

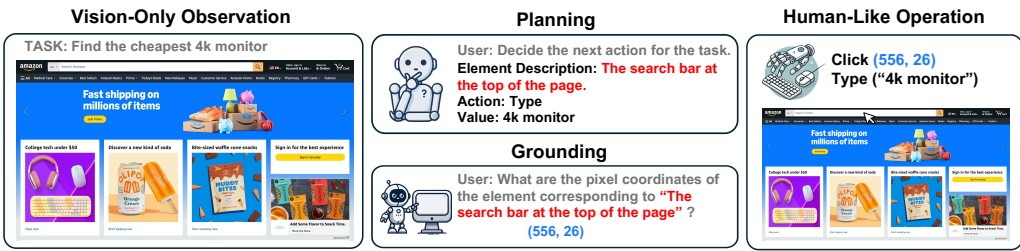

Figure 2: **SeeAct-V**, which uses screenshots as the only environmental observation (task instructions are input as text), without relying on HTML or a11y trees. It includes an MLLM that generates textual plans and a visual grounding model to map textual plans into coordinates on the screenshot. *Note: "Click" is always automatically inserted before "Type."*

### 2.1 OVERVIEW

We adapt the popular SeeAct framework (Zheng et al., 2024) to one in which agents only take visual observation of the environment and directly conduct pixel-level operations, denoted as *SeeAct-V* (Figure 2). The original SeeAct has two stages: planning and grounding, both handled by an MLLM. At each step, the MLLM first generates a textual plan, then selects grounding candidates from a short list. The grounding candidates are either filtered HTML elements or labels of Set-of-Mark (SoM; Yang et al. (2023)) annotations on the screenshot, both of which require HTMLs or a11y trees as additional input. In contrast, SeeAct-V only uses screenshots for environmental observation. For grounding, SeeAct-V uses a separate model specialized for visual grounding that directly produces the coordinates on the current screen where the agent should act. We provide our philosophy behind the modular design of SeeAct-V in Appendix B.

A strong visual grounding model therefore becomes the key for making SeeAct-V a compelling framework. Ideally, it should generalize across platforms (e.g., web, desktop, and mobile) and handle diverse ways of referring to GUI elements. Considering the rapid evolution of MLLMs, this grounding model should be easily pluggable into different MLLMs to help ground their plans into different GUI environments. Finally, GUI screenshots can vary drastically in resolution and orientation, therefore the grounding model should handle a wide range of input resolutions. *The main technical contribution of this work is a surprisingly simple recipe (incl. data and modeling) for training such universal visual grounding models.* We introduce our simple data synthesis strategy in §2.2, followed by modeling considerations in §2.3. With this simple recipe, we construct the largest training data for GUI grounding to date and train *UGround*, a strong universal visual grounding model for GUI agents.

## 2.2 DATA CONSTRUCTION

We synthesize a large, high-quality, and diverse set of ⟨*screenshot*, *referring expression*, *coordinates*⟩ triplets as training data for visual grounding, where we use the center point coordinates of an element as the expected output. Our data synthesis is fully based on webpages. Webpages are ideal for grounding data synthesis because of their *dual* representation—we can easily get the full HTML, the visual rendering, and fine-grained correspondences between the two (e.g., HTML elements to precise bounding boxes). HTML elements also contain rich metadata such as CSS or accessibility attributes, opening numerous opportunities for synthesizing diverse referring expressions (REs). Finally, since GUI designs share many similarities across platforms, we hypothesize that visual grounding models trained only on web data will generalize to other platforms like desktop and mobile UIs.

**Common RE Types for GUIs**. People use diverse ways to refer to GUI elements (Figure 3). Previous visual grounding works (Hong et al., 2024; Cheng et al., 2024) have not sufficiently considered this dimension of diversity. We categorize common REs for GUI elements into three types: 1) **Visual REs**, i.e., salient visual features like text or image content, element types (e.g., buttons or input fields), shapes, colors, etc. 2) **Positional REs**, including both absolute (e.g., "*at the top left of the page*") and relative positions (e.g., "*to the right of element X*") to other elements. Besides straightforward positional information, *contextual references* (e.g., "*for Item A*," "*under the section X*") are more challenging for grounding because they require understanding both positional relationships and semantic relationships between elements (e.g., a like button is associated with a product). 3) **Functional REs**, i.e., referring to elements by their main functions (e.g., "*Navigate to Home*," "*Go to My Cart*"). Composite types that combine two or more of these types are also common, especially when stronger disambiguation is needed, e.g., "*click the heart button under the Pokémon shirt to add to favorite*."

Figure 3: Examples of visual, positional, and functional REs.

**Hybrid RE Synthesis from Web**. We propose a novel hybrid synthesis pipeline, orchestrating both carefully curated rules as well as LLMs to generate diverse REs for HTML elements: 1) **Primary Descriptors:** We extract abundant visual and functional information that are embedded in the attributes of HTML elements. For example, HTML attributes like `inner-text` and `alt` provide visual clues (including text content), while accessibility attributes like `aria-label` reveal more functional aspects of an HTML element. However, HTML attributes are often incomplete. To harvest visual and functional signals beyond HTML attributes, we use an open MLLM, LLaVA-NeXT-13B (Liu et al., 2024b). We input the visual rendering of an HTML element along with its available attributes to the MLLM and prompt it to generate diverse REs. This process often yields composite REs that combine some HTML attributes with visual features (e.g., "*hollow heart*") or new knowledge from the MLLM (e.g., a blue bird icon represents Twitter). Similar to Lai et al. (2023), we also employ an LLM (Llama-3-8B-Instruct; AI@Meta (2024)) to make these generated REs more concise. We randomly select an HTML attribute (that may contain functional or visual information) or the synthesized description by LLMs as the *primary descriptor* of an element. 2) **Positional Expressions:** We curate rules to generate positional REs according to the absolute position of an element in the screenshot as well as its spatial relationship to neighboring elements (e.g., "*at the top of the page*," "*between element A and B*"). We also create multiple rules to generate contextual references. For example, we identify elements of certain types in the screenshot (e.g., radio buttons, checkboxes, input fields), and generate REs for them based on their spatial and structural relationship (e.g., hierarchical structure of the DOM tree) to others (e.g., "*the input field labeled Birthday*").

We collect screenshots (mix of portrait and landscape views in various resolutions) and metadata of web elements (salient HTML attributes, bounding box coordinates) from Common Crawl,[4] and then

---

[4] https://commoncrawl.org/

Table 1: Overview of training datasets used for UGround.

| Dataset | Annotation | # of Elements | # of Screenshots | Platform |
|---|---|---|---|---|
| Web-Hybrid (Ours) | Rule + LLM | 9M | 773K | Web |
| Web-Direct (Ours) | GPT | 408K | 408K | Web |
| GUIAct (Chen et al., 2024) | GPT + Human | 140K | 13K | Web |
| AndroidControl (Li et al., 2024b) | Human | 47K | 47K | Android |
| Widget Caption (Li et al., 2020b) | Human | 41K | 15K | Android |
| UIBert (Bai et al., 2021) | Human | 16K | 5K | Android |
| AITZ (Zhang et al., 2024c) | GPT + Human | 8K | 8K | Android |
| Total | | 10M | 1.3M | Web + Android |

apply our data synthesis pipeline to get our main training dataset (**Web-Hybrid**). We leave more details to Appendix E.1.

**Supplementary Data**. There have been multiple prior efforts on constructing grounding data for Android, so we incorporate the existing datasets as well. We also use GPT-4o to directly synthesize a small set of REs for web elements, with a focus on more open-ended REs (no constraints on the type) and functional REs (**Web-Direct**). These additions help provide more diverse REs and cover elements in Android, especially those not commonly found on the web (e.g., toggle buttons).

In total, we compile a dataset totaling 10M UI elements, with the majority (90%) from our hybrid synthesis pipeline (Table 1). Elements on the same screenshot are batched to accelerate training.

## 2.3 MODEL DESIGN

We adopt a widely used open-source model architecture, 7B LLaVA-NeXT (Liu et al., 2024b), as our backbone model for visual grounding. We make a few adaptations to tailor it for GUI grounding.

**Input-Output Formulation**. We always instruct the model to answer "*In the screenshot, what are the pixel element coordinates corresponding to {Description}?*" Following recent work in visual grounding (Cheng et al., 2024), we represent the answer in natural language so we can directly use autoregressive decoding. Specifically, we opt for coordinates in the numerical form (e.g., "*(1344, 1344)*") to precisely point to an element without any normalization.

**Image Resolution**. GUI screenshots are much larger than typical natural images, often requiring a resolution above 1,000px for legibility. LLaVA (Liu et al., 2024c;a) was initially built for 336px images, and was later scaled up to at most 772px via the AnyRes technique (Cheng et al., 2023; Gao et al., 2024; Liu et al., 2024b; Guo et al., 2024; Dong et al., 2024). It resizes and splits a large image into small slices, encodes each slice independently with the vision encoder, and adds a special token at the end of each row to help the language model keep track of the image shape. AnyRes allows easy scaling up of input resolution. However, it is always a trade-off between the diversity of supported resolutions and the speed of training and inference. To strike a balance and avoid meaningless excessive resolutions, we enlarge the allowed input sizes to 36 ViT (Dosovitskiy et al., 2021) slices, and use CLIP@224px (Radford et al., 2021) as the image encoder for more flexible splitting, pushing the maximum supported resolution to $1,344 \times 1,344$ (landscape) and $896 \times 2,016$ (portrait). Additionally, we use Vicuna-1.5-7b-16k (Zheng et al., 2023) with 16K context length to handle long visual contexts. Finally, there is a low-resolution image fusion module commonly used in AnyRes. However, we find it ineffective for GUI grounding, as 224px is too small to provide informative global context, so we leave it out from our model. More details are in Appendix F.

## 3 EXPERIMENTS

Most existing studies on GUI agents typically evaluate on one or two benchmarks. In contrast, we conduct a much more comprehensive evaluation on GUI agents to show the universality of our method. Our evaluation employs six benchmarks that span all three major platforms (i.e., web, desktop, and mobile) and cover three settings: *visual grounding* (§3.1), *offline agent evaluation* on cached environment states (§3.2), and *online agent evaluation* in live environments (§3.3). The visual grounding setting focuses on the grounding performance of UGround, while the agent settings test the end-to-end effectiveness of the SeeAct-V framework with UGround integrated. On the agent

Table 2: Grounding accuracy on ScreenSpot (Standard Setting). Results for GPT-4, CogAgent, and SeeClick are from Cheng et al. (2024).

| Grounding Model | Mobile | | Desktop | | Web | | Average |
|---|---|---|---|---|---|---|---|
| | Text | Icon/Widget | Text | Icon/Widget | Text | Icon/Widget | |
| GPT-4 | 22.6 | 24.5 | 20.2 | 11.8 | 9.2 | 8.8 | 16.2 |
| GPT-4o | 20.2 | 24.9 | 21.1 | 23.6 | 12.2 | 7.8 | 18.3 |
| CogAgent (Hong et al., 2024) | 67.0 | 24.0 | 74.2 | 20.0 | 70.4 | 28.6 | 47.4 |
| SeeClick (Cheng et al., 2024) | 78.0 | 52.0 | 72.2 | 30.0 | 55.7 | 32.5 | 53.4 |
| UGround (Ours) | **82.8** | **60.3** | **82.5** | **63.6** | **80.4** | **70.4** | **73.3** |

Table 3: Grounding accuracy on ScreenSpot (Agent Setting) with planner-generated REs.

| Planner | Grounding | Mobile | | Desktop | | Web | | Avg. |
|---|---|---|---|---|---|---|---|---|
| | | Text | Icon/Widget | Text | Icon/Widget | Text | Icon/Widget | |
| GPT-4 | SeeClick | 76.6 | 55.5 | 68.0 | 28.6 | 40.9 | 23.3 | 48.8 |
| | UGround | 90.1 | 70.3 | 87.1 | 55.7 | 85.7 | 64.6 | 75.6 |
| GPT-4o | SeeClick | 81.0 | 59.8 | 69.6 | 33.6 | 43.9 | 26.2 | 52.3 |
| | UGround | **93.4** | **76.9** | **92.8** | **67.9** | **88.7** | **68.9** | **81.4** |

benchmarks, we compare the vision-only SeeAct-V framework with prior SOTA methods that usually require additional text-based representations (HTML or a11y tree) as input. Within SeeAct-V, we also compare UGround with existing visual grounding models whenever possible.

## 3.1 GUI VISUAL GROUNDING

We first evaluate UGround on the ScreenSpot benchmark (Cheng et al., 2024), which is specifically designed for visual grounding on GUIs. The benchmark consists of 1,272 single-step instructions and the corresponding bounding boxes of the target elements across mobile (e.g., iOS and Android), desktop (e.g., macOS and Windows), and web environments. These elements vary between text-based elements, icons (e.g., the trash can icon) and widgets (e.g., to-do lists), representing diverse GUI element types.

We evaluate under two settings: 1) **Standard Setting**. In the standard setting of ScreenSpot, the instructions are written by human annotators with a primary focus on *functional description* of the target elements, e.g., simply "*close*" to refer to the 'X' button that closes a window or "*set an alarm for 7:40*" when the input image shows the iPhone clock app with a list of inactive alarms. 2) **Agent Setting**. For GUI agents, a grounding model needs to work with a planning model (e.g., an MLLM) and ground the REs it generates, which includes not only functional REs but also visual and positional REs (see §2.2). To provide a more comprehensive evaluation on visual grounding for GUI agents, we input each ScreenSpot example to an MLLM, which acts as a planning model, and asks it to generate diverse REs for the target element. This setting is therefore more representative of the grounding challenges in GUI agents. We mainly compare UGround with SeeClick (Cheng et al., 2024), the state-of-the-art visual grounding model on ScreenSpot, and another visual grounding model CogAgent (Hong et al., 2024). To show the challenge of visual grounding for general-purpose models, we also compare with GPT-4 and GPT-4o.

**Results**. As shown in Table 2 and Table 3, UGround outperforms all existing models across all the settings and platforms by a substantial margin, about an absolute improvement of **20%** on average under the standard setting and **29%** under the agent setting. Interestingly, UGround performs remarkably well on desktop UIs, despite the fact that it is never trained on desktop screenshots (Table 1). Compared with existing models, UGround performs especially well on icons and widgets, which are generally more challenging for grounding because that requires deeper understanding of the contextual (e.g., positional) and semantic (e.g., functional) information. Overall, the strong results on ScreenSpot clearly demonstrates UGround's universal grounding capability across platforms and planners as well as the remarkable effectiveness of our simple data synthesis and modeling recipe.

Table 4: Element accuracy on Multimodal-Mind2Web. Results by Choice and SoM are from Zheng et al. (2024). The SoM results are on subsets of 30 tasks for each split.

| Input | Planner | Grounding | Cross-Task | Cross-Website | Cross-Domain | Avg. |
|---|---|---|---|---|---|---|
| Image + Text | GPT-4 | Choice | 46.4 | 38.0 | 42.4 | 42.3 |
| | | SoM | 29.6 | 20.1 | 27.0 | 25.6 |
| Image (SeeAct-V) | GPT-4 | SeeClick | 29.7 | 28.5 | 30.7 | 29.6 |
| | | UGround | 45.1 | 44.7 | 44.6 | 44.8 |
| | GPT-4o | SeeClick | 32.1 | 33.1 | 33.5 | 32.9 |
| | | UGround | **47.7** | **46.0** | **46.6** | **46.8** |

## 3.2 OFFLINE AGENT EVALUATION

We discuss the experimental setup for three offline agent evaluation benchmarks followed by result discussion. Concrete examples from each benchmark are given in Appendix D.

**Web: Multimodal-Mind2Web.** We use Multimodal-Mind2Web (Zheng et al., 2024), the multimodal extension of Mind2Web (Deng et al., 2023), for our evaluation on realistic web tasks. The test split consists of 1,013 tasks spanning over 100 different websites. Each task contains a high-level task instruction and a sequence of actions, with a screenshot of the webpage before each action, as the golden trajectory. All the webpages along the golden trajectory are cached to support offline evaluation. The tasks are crowdsourced with a focus on ensuring real-world meaningfulness (i.e., what real users would need on those websites).

Zheng et al. (2024) have clearly demonstrated the necessity of visual perception for web agents, so we mainly compare with zero-shot methods that use MLLMs as planners and omit text-only LLMs. Zheng et al. (2024) have also identified grounding as the main challenge and proposed several grounding strategies, including 1) **Choice**, where the planner is asked to choose from a short list of filtered HTML elements, and 2) **SoM**, where the input screenshot is superposed with Set-of-Mark (Yang et al., 2023) labels and the planner is asked to select from the labels. Both strategies require additional text-based representations (i.e., HTML) to obtain the candidates and/or locate the elements in the screenshot to label. We report *element accuracy*, i.e., accuracy of selecting the correct element, and omit operation scores because they are orthogonal to grounding comparisons.

**Mobile: AndroidControl.** We use AndroidControl (Li et al., 2024b), a large-scale Android dataset comprising 15K unique tasks over 833 Apps. Screenshots, action sequences, and a11y trees are cached from human demonstrations as golden trajectories for training and evaluation purposes. Each action is also labeled by a corresponding low-level instruction (e.g., "*set the hours to 6*"). Following Li et al. (2024b), we use 500 random steps from the test set. We compare with the SOTA zero-shot method, the text-only version of M3A (Rawles et al., 2024), which instructs GPT-4 to generate textual actions as well as select elements from the a11y tree (**Choice**). We adopt the two task settings in Li et al. (2024b): high-level tasks, where only the high-level intent is provided, and low-level tasks, where both the high-level intent and the corresponding low-level instruction for each step are available. We use the standard metric, *step-wise accuracy*, where a step is considered successful only if all the predicted actions, elements, and arguments (if applicable) are correct.

**Desktop: OmniACT**. We use OmniACT (Kapoor et al., 2024) to evaluate the accuracy of UGround on desktop tasks. The dataset consists of 9,802 tasks covering 38 desktop applications and 27 websites across different desktop platforms (macOS, Windows, and Linux). Each task requires the generation of a PyAutoGUI script, which is a sequence of actions to complete the task on a single screenshot. The SOTA method, DetACT (Kapoor et al., 2024), extracts UI elements and their coordinates through a combination of OCR (optical character recognition), icon matching, and color detection modules. These elements are filtered by task relevance and then passed to LLMs or MLLMs to generate the PyAutoGUI script with the appropriate coordinates for interaction.

For SeeAct-V, we replace the input of the DetACT pipeline with only screenshots and instruct MLLMs to generate element descriptions rather than directly generate coordinates. We then employ UGround to obtain the coordinates of the elements, which are subsequently integrated into the PyAutoGUI scripts. To ensure a fair comparison, we strictly follow the approach in Kapoor et al. (2024), including the same prompt and retrieval strategy that selects five in-context examples from the training set based on task similarity. We report the *action score*, which measures the accuracy of the action sequences while penalizing errors in generated arguments.

Table 5: Step accuracy on AndroidControl over 500 random actions from the test split. Baseline results are from Li et al. (2024b).

| Input | Planner | Grounding | Step Accuracy | |
|---|---|---|---|---|
| | | | **High** | **Low** |
| Text | GPT-4 | Choice | 42.1 | 55.0 |
| Image (SeeAct-V) | GPT-4 | SeeClick | 39.4 | 47.2 |
| | | UGround | 46.2 | 58.0 |
| | GPT-4o | SeeClick | 41.8 | 52.8 |
| | | UGround | **48.4** | **62.4** |

Table 6: Action scores (AS) on OmniACT. Baseline results are from Kapoor et al. (2024).

| Inputs | Planner | Grounding | AS |
|---|---|---|---|
| Text | GPT-4 | DetACT | 11.6 |
| Image + Text | | DetACT | 17.0 |
| Image (SeeAct-V) | GPT-4 | SeeClick | 28.9 |
| | | UGround | 31.1 |
| | GPT-4o | SeeClick | 29.6 |
| | | UGround | **32.8** |

Table 7: Completion rate (CR) and task success rate (SR) on Mind2Web-Live. Baseline results are from Pan et al. (2024).

| Inputs | Planner | Grounding | CR | SR |
|---|---|---|---|---|
| Text | GPT-4 | Choice | 44.3 | 21.1 |
| | GPT-4o | | 47.6 | 22.1 |
| Image (SeeAct-V) | GPT-4 | UGround | 50.7 | **23.1** |
| | GPT-4o | | **50.8** | 19.2 |

Table 8: Task success rate (SR) on Android-World. Baseline results are from Rawles et al. (2024).

| Input | Planner | Grounding | SR |
|---|---|---|---|
| Text | GPT-4 | Choice | 30.6 |
| Image + Text | | SoM | 25.4 |
| Image (SeeAct-V) | GPT-4 | UGround | 31.0 |
| | GPT-4o | | **32.8** |

**Results**. As shown in Table 4, Table 5, and Table 6, SeeAct-V with UGround outperforms all the baselines across the board, despite only using raw screenshots as input while baselines use additional input. UGround also consistently outperforms a strong GUI grounding model, SeeClick. These results provide solid support for human-like vision-only embodiment for GUI agents, a position this work aims to make a case for. The results also further validate UGround's efficacy as a universal grounding model for GUI agents.

## 3.3 ONLINE AGENT EVALUATION

We further evaluate our approach in an end-to-end manner on two online agent benchmarks that closely resemble the offline web and Android benchmarks in §3.2, but involve interactions with live websites and mobile applications. Due to the high cost of online evaluation, we only use UGround for grounding.

**Web: Mind2Web-Live**. We use the test set from Mind2Web-Live (Pan et al., 2024). The benchmark is built on Mind2Web (Deng et al., 2023) by adding functional evaluation to the tasks that makes automated evaluation possible on live websites. Specifically, it defines and annotates *key nodes* for each task, which are critical steps that must be completed for a task to be considered successful, regardless of which trajectory an agent takes. The baseline agent from Pan et al. (2024) is text-only, perceives and interacts with webpages by hundreds of HTML elements at a time. For SeeAct-V, we change the observation to be screenshots only, and make necessary changes to the original action space to fully eliminate the dependency on HTML during planning, grounding, and execution (details in Appendix G.5). We use standard metrics: *micro completion rate*, which measures the proportion of completed key nodes across all the tasks, and *task success rate*, which measures the proportion of fully completed tasks.

**Mobile: AndroidWorld.** We use AndroidWorld (Rawles et al., 2024), an online mobile agent benchmark running in Android emulators. It includes 116 tasks across 20 Apps, with evaluation based on the final states of the device. We compare with the SOTA agent M3A and its text-only variant from Rawles et al. (2024). They receives both raw and SoM images, together with textual UI elements, or only the textual UI elements as the observation respectively. Both variants employ a ReAct-style reasoning process (Yao et al., 2023) to select the next target element from a list of UI elements. Additionally, they integrate self-reflection (Shinn et al., 2024) for the agent to summarize its current action and improve decision-making in subsequent steps. We report *task success rate*, which measures the percentage of fully completed tasks.

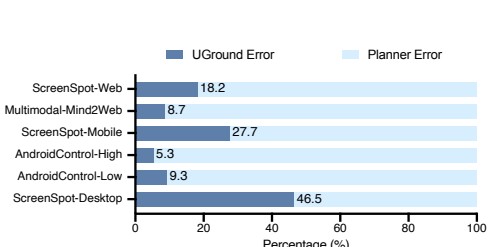

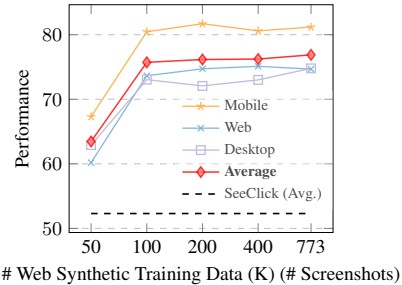

Figure 4: Error distribution from manual analysis.

Figure 5: Scaling curve of UGround on ScreenSpot w.r.t. Web-Hybrid data size.

**Results**. SeeAct-V with UGround gets comparable or higher performance in online agent evaluation, as shown in Table 7 and Table 8. Particularly, it achieves a much higher success rate compared with the SoM variant of M3A, even though Android environments have less dense UI layouts and are generally more suitable for SoM (i.e., less obstruction by the SoM labels). These results again provide solid support for the feasibility and promises of human-like vision-only embodiment for GUI agents and the effectiveness of UGround.

## 3.4 Error Analysis

We conduct a manual error analysis of the best performing method, SeeAct-V with UGround, to understand the bottleneck for further improvement. We randomly sample 60 failure cases from each split of ScreenSpot (agent setting with GPT-4o), AndroidControl, and Multimodal-Mind2Web. Except for data annotation errors, errors from the models can be categorized into *planning errors*, i.e., generating plans with incorrect element descriptions, and *grounding errors*, i.e., predicting incorrect coordinates for a correct element description from the planner.

As shown in Figure 4, planning errors are the dominant cause of failures across all benchmarks, further confirming the strong grounding capability of UGround. The most frequent error is that the planner generates (otherwise correct) description of an incorrect element on the screen, indicating a lack of correct understanding of either the task and/or the elements. Other common planning errors include hallucinating non-existent elements or producing overly generic descriptions that are too vague to uniquely locate the target element, even for human evaluators.

On the other hand, on ScreenSpot-Mobile and ScreenSpot-Desktop, a considerable portion of the failures do stem from grounding errors. Both desktop and mobile UIs feature a pervasive use of icons with idiosyncratic meaning. For example, a stylized dollar sign represents the Zelle App, or an icon with two cartoon people represents one's contact list in Microsoft Outlook. We find that pretrained MLLMs and our web-centric grounding training are effective in capturing the semantics of popular icons (e.g., icons representing Google) or commonsense meaning (e.g., clock icons usually represent time-related functions like alarms). However, it is challenging to capture the idiosyncratic semantics of icons in the long tail, which arguably requires either additional documentation or more targeted exploration to learn. This is a major cause of the grounding errors. Interestingly, when tested on more realistic agent tasks, e.g., in AndroidControl, AndroidWorld, and OmniACT, UGround still proves to be relatively robust. This is because most of the agent tasks concern things in the head of the distribution; things in the long tail are naturally rare (though still important). This explains the strong performance of UGround on mobile and desktop agent benchmarks. Nonetheless, how to capture idiosyncratic semantics in the long tail is still an open challenge for grounding.

## 3.5 Training Data Analysis: Scaling and Ablations

We conduct scaling analysis and ablation studies on our training data to better understand the contribution of different data for UGround's strong performance, and use the agent setting of ScreenSpot for the evaluation (with GPT-4o as the planner). Further ablations around data, model design, and RE types are provided in Appendix C.

Table 9: Training data ablations for UGround on ScreenSpot (Agent Setting).

| Training Data | Mobile | | Desktop | | Web | | Average |
|---|---|---|---|---|---|---|---|
| | Text | Icon/Widget | Text | Icon/Widget | Text | Icon/Widget | |
| Web-Hybrid | 89.0 | **73.4** | **88.1** | **61.4** | 84.8 | **64.6** | **76.9** |
| Others | **92.3** | 71.2 | 84.5 | 46.4 | **87.0** | 59.2 | 73.4 |
| All | 93.4 | 76.9 | 92.8 | 67.9 | 88.7 | 68.9 | 81.4 |

**Scaling Curve on Web-Hybrid**. We investigate the scaling of our primary synthetic dataset, Web-Hybrid, which consists of 9M data instances over 773K web screenshots in total. The scaling results in Figure 5 show that the average performance consistently improves as the data scales up, though the return starts diminishing after 100K screenshots. Notably, with just 50K screenshots (about 600K elements) as training data, UGround surpasses SeeClick by more than 10%, which is trained on about 3M web and Android elements from about 400K screenshots. The results clearly show the high data quality and the effectiveness for grounding training of our data synthesis pipeline. Upon manual inspection, we observe that additional data after 100K screenshots primarily enhances understanding of less frequent elements such as radio buttons, checkboxes, or very small text elements. As data increases, the model can point to the center of element bounding boxes more accurately and better handle tiny hyperlinks.

**Training Data Ablations**. To further investigate the impact of training data sources, we compare the performance of UGround trained on only Web-Hybrid, only the supplementary data, or both (see Table 1). Results in Table 9 further validate the necessity of Web-Hybrid. Training on other data without Web-Hybrid often underperforms training on Web-Hybrid alone. This is most evident on icons and widgets, which require understanding more diverse aspects, such as visual features and functions, than text-based elements. Finally, these two data sources are complementary and their combination yield the best performance across the board.

## 4 CONCLUSIONS AND LIMITATIONS

We introduce UGround, a universal GUI visual grounding model developed with large-scale web-based synthetic data. UGround shows strong cross-platform generalization and substantially outperforms the prior models. We propose a vision-only framework SeeAct-V that allows pixel-level interactions based solely on visual input. Comprehensive evaluation on both offline and online agent benchmarks demonstrates that SeeAct-V agents with UGround can achieve comparable and often better performance than prior SOTA agents that rely on additional textual inputs like HTML or a11y trees for observation or grounding.

Nevertheless, there are still some limitations that could be addressed in future work to advance visual grounding in GUI applications and visually grounded GUI agents. First, UGround is trained on very large-scale synthetic data. Considering the similarity and repetition of elements between web pages, there is room to improve on data efficiency during training, for example by better data grouping and deduplication. On the other hand, despite the cross-platform generalization shown in our experiment results, the issue of long-tail elements remains under-addressed in this work. Mobile UIs and desktop UIs often feature specific icons with idiosyncratic semantics, and it can be impractical to account for every long-tail element in a training set. Additionally, no desktop UI data is incorporated in the training of this work, which limits the performance on desktop UIs. Given the scarcity of training datasets for desktop UIs, we anticipate the development of more comprehensive datasets in this domain. Lastly, UGround depends on an external planner; it is not meant to function independently as a GUI agent. Nonetheless, we hope that our datasets, model, and framework can contribute to future studies of vision-only agents, as well as contribute to advancing the grounding capabilities of end-to-end models, as strong grounding data has been shown to improve end-to-end models (Cheng et al., 2024; Hong et al., 2024; Chen et al., 2024).

## ETHICS STATEMENT

This work employs web-based data synthesis to develop visual grounding models for GUIs. The synthesis pipeline and data collection presented in this paper are intended solely for research purposes

related to GUI grounding and GUI agents, in line with prior works in the field (Hong et al., 2024; Cheng et al., 2024).

The webpages utilized in our work are sourced from the Common Crawl dataset[5], which is a publicly available Internet archive for research and non-commercial use. We use only a small subset of it and strictly adhere to Common Crawl's terms of use[6] throughout our study.

Our use and dissemination of the data are exclusively for academic research and fully comply with Section 107 of the U.S. Copyright Law regarding Fair Use. Prior to release, the data undergoes rigorous content moderation. We acknowledge full responsibility for any legal issues arising from our data collection and accept all associated risks. Furthermore, the distribution of the data is managed in strict accordance with applicable regulations and guidelines to ensure compliance with AI ethics standards and non-commercial usage.

## ACKNOWLEDGMENTS

We are grateful for the collaboration with the Orby AI team (particularly Sanjari Srivastava, Peng Qi, Gang Li, and Will Lu) for their contribution on data collection and analysis, as well as for providing computing resources. We would also like to extend our appreciation to colleagues from the OSU NLP group and Kanzhi Cheng, Yulu Guo, Lizi Yang for their insightful comments. Special thanks to Yichen Pan, Christopher Rawles, Dehan Kong, Alice Li, and Raghav Kapoor for their assistance with evaluation. This work is supported in part by Orby AI, ARL W911NF2220144, and NSF CAREER #1942980. The views and conclusions contained herein are those of the authors and should not be interpreted as representing the official policies, either expressed or implied, of the U.S. government. The U.S. government is authorized to reproduce and distribute reprints for government purposes notwithstanding any copyright notice herein.

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

# Table of Contents in Appendix

# A    RELATED WORK

**GUI Agents.** LLMs and MLLMs have demonstrated great capabilities and potentials in GUI automation, working as digital agents in various GUI environments (Yan et al., 2023; Kim et al., 2023; Wang et al., 2024a; Zheng et al., 2024; Xie et al., 2024). Despite the growing number of studies focused on building multimodal agents (Koh et al., 2024; Zhou et al., 2024; Cao et al., 2024), most work still relies on HTML or a11y trees for grounding, even when they are not used for observation. In this work, we advance an alternative line of research: pixel-level visually grounded GUI agents (Shaw et al., 2023; Zhan & Zhang, 2023; Hong et al., 2024; Cheng et al., 2024; Niu et al., 2024). Unlike nearly all previous work of this line, we propose a generic two-stage approach that separates planning and visual grounding to build vision-only GUI agents, which perform remarkably well on realistic agent benchmarks with vision-only input, and offers the flexibility to the choices of planning and grounding models.

**Visual Grounding.** Visual grounding has been long studied on natural images (Karpathy et al., 2014; Mao et al., 2016; Yu et al., 2016). More recently, with the advancements of MLLMs, their visual grounding capabilities on natural images have attracted significant attention (Bai et al., 2023; Chen et al., 2023a;b; Peng et al., 2023; Wang et al., 2024b; 2023; Ma et al., 2024). However, due to significant gaps in image resolution and GUI understanding, these models trained on natural contexts work poorly on GUI visual grounding (Cheng et al., 2024). One of the most popular approaches, SoM (Yang et al., 2023), proposes a visual prompting method that adds marks such as boxes and numbers to images and instructs MLLMs to identify the referred objects by the labels. It is widely adopted in GUI scenarios (Yan et al., 2023; He et al., 2024; Koh et al., 2024), but still suffers from problems including reliance on complete object information or object segmentation. Only few studies have been conducted for visual grounding on GUI screenshots. Based on Rico (Deka et al., 2017), Bai et al. (2021) annotates referring expressions by humans; RicoSCA (Li et al., 2020a) generates a larger synthetic referring expression dataset; and Li et al. (2020b) collect human-labeled captions of UI elements. They have been primary resources for GUI grounding for a long time (Li & Li, 2022; Banerjee et al., 2022). Later on, Qian et al. (2024) synthesize referring expressions from Rico by heuristic rules and train a vision language model by a new layout-aware contrastive learning technique. CogAgent (Hong et al., 2024) compiles HTML documents and screenshots from real websites to GUI grounding data for the pretraining stage, and finetunes on open-source and in-house human-labeled data, to build a 18B MLLM with strong pixel-level GUI grounding capabilities. Ferret-UI (You et al., 2024) develops a UI generalist MLLM trained on a series of UI-related tasks including grounding. The most similar effort to ours is SeeClick (Cheng et al., 2024), which enhances Qwen-VL (Bai et al., 2023) by finetuning on GUI grounding data, including simplistic synthetic data compiled from real websites. It still falls short of the small image resolution of Qwen-VL, as well as the simplistic nature of the training data. Cheng et al. (2024) also create a new grounding benchmark for GUIs, which benefits our evaluation and analysis.

## B  PHILOSOPHY BEHIND SEEACT-V AND UGROUND

When it comes to agent designs, the current wisdom, by and large, is to train a monolithic LLM (e.g., CogAgent (Hong et al., 2024), SeeClick (Cheng et al., 2024), along with several recent supervised fine-tuning endeavors aimed at enhancing *"agentic behaviors"*). At a philosophical level, part of the goal of SeeAct-V is to challenge that *status quo* and advocate a modular design for language agents instead.

A fundamental challenge of language agents arises from the complexity, dynamism, and inherent idiosyncrasies of the environments in which they operate. For instance, consider web agents: the internet comprises over one billion websites, each of which can exhibit an extremely large and dynamic number of states, and each can be constantly changing (for example, due to frequent updates in backend databases). Furthermore, there is a considerable amount of highly idiosyncratic semantics in each environment, e.g., uncommon icons, jargon, and counter-intuitive designs.

As a result, although we are still at the early stage of agent research, we posit that a monolithic model, regardless of its future scale and capabilities, is unlikely to fully encapsulate the diverse complexities and idiosyncrasies across all environments. Therefore, developing a generalist agent that reliably generalizes across various contexts necessitates a modular system design. This involves synergistically orchestrating a foundation model (e.g., GPT-4o) with multiple specialized modules, each tailored to specific functionalities.

Grounding, in particular, is a capability for which a dedicated module is highly advantageous. Fundamentally, grounding involves interpreting domain-specific semantics and creating a map between that and natural language representations understood by a generic LLM. A specialized grounding module simplifies the capture of idiosyncratic semantics and facilitates easier adaptation across different domains (for example, by fine-tuning the grounding model rather than the entire foundation model). Consequently, the grounding module provides domain-specific semantic input to the foundation model. This constitutes a central motivation for the design of SeeAct-V and the work presented herein.

Our design also offers several practical advantages:

**Modularity**: It permits the independent study and enhancement of UGround as a standalone grounding model, decoupled from specific planning modules.

**Flexibility**: It is compatible with diverse multimodal LLMs and grounding models without requiring specialized fine-tuning on downstream benchmarks.

**Comparative Consistency**: By standardizing the planning stage, the design minimizes confounding variables, thereby facilitating a clearer assessment of how various grounding models and methods influence agent performance.

Empirical results demonstrate that SeeAct-V, when integrated with UGround, outperforms end-to-end MLLMs (whether employing textual or SoM grounding). This is particularly noteworthy considering that training end-to-end models demands extensive high-quality data on agent trajectories (which combine both planning and grounding), which is both challenging and costly.

## C  FURTHER ABLATION STUDIES

In addition to the studies in §3.5, we present further ablation experiments to investigate both model design choices and the effectiveness of our web-based synthetic dataset. We report grounding accuracy on ScreenSpot (Agent Setting), with GPT-4o as the planner.

### C.1  CONTROLLED COMPARISON TO BASELINE MODELS

Both model design and training data contribute critically to the strong performance of UGround. To isolate their individual contributions, we introduce a new variant, UGround-Qwen, which is fine-tuned

Table C.1: Ablations of data and base models for UGround on ScreenSpot (Agent Setting).

| Model | Model Design | Continual SFT Data | Mobile | | Desktop | | Web | | Avg |
|---|---|---|---|---|---|---|---|---|---|
| | | | Text | Icon/Widget | Text | Icon/Widget | Text | Icon/Widget | |
| Qwen-VL-Chat | Qwen-VL | None | 21.3 | 21.4 | 18.6 | 10.7 | 9.1 | 5.8 | 14.5 |
| SeeClick | Qwen-VL | Full SeeClick | **81.0** | **59.8** | 69.6 | 33.6 | 43.9 | 26.2 | 52.3 |
| UGround-Qwen | Qwen-VL | Web-Hybrid | 80.2 | 57.2 | 76.3 | **39.3** | 74.4 | 47.1 | 62.4 |
| UGround | Ours | Web-Hybrid | **89.0** | **73.4** | **88.1** | **61.4** | **84.8** | **64.6** | **76.9** |

Table C.2: Ablations of image resolution for UGround on ScreenSpot (Agent Setting).

| Continual SFT Data | Image Resolution | Mobile | | Desktop | | Web | | Avg. |
|---|---|---|---|---|---|---|---|---|
| | | Text | Icon/Widget | Text | Icon/Widget | Text | Icon/Widget | |
| | Fixed 448 x 448 | **89.4** | 65.1 | 83.5 | 56.4 | 77.0 | 61.7 | 72.2 |
| Web-Hybrid | Fixed 896 x 896 | 86.8 | 69.0 | 85.1 | **62.9** | 81.4 | 57.8 | 73.8 |
| | Fixed 1,344 x 1,344 | 79.9 | 68.6 | 86.1 | 62.1 | 79.1 | 63.6 | 73.2 |
| | Dynamic (Ours) | 89.0 | **73.4** | **88.1** | 61.4 | **84.8** | **64.6** | **76.9** |

from Qwen-VL-Chat (the same backbone used in SeeClick), using only our main web-based synthetic dataset, Web-Hybrid[7]. The results are presented in Table C.1.

**Training Data:** When using the same backbone (Qwen-VL-Chat), UGround-Qwen trained solely on Web-Hybrid achieves an average absolute improvement of 10.1% over SeeClick, even though SeeClick incorporates additional open-source mobile UI data. This result underscores both the high quality of our synthetic web data and its capability to generalize across platforms.

**Model Design:** UGround demonstrates a 14.5% absolute improvement over UGround-Qwen, thereby highlighting the effectiveness of our model design.

We omit comparisons with CogAgent due to its inferior performance relative to SeeClick, despite its substantially larger model size (18B parameters) and dataset (140M grounding samples).

## C.2   MODEL DESIGN

We analyze the effect of image resolution on performance, focusing on two key aspects: (1) the impact of increasing image resolution using scaled-up `AnyRes` grid settings, and (2) the benefits of dynamic resolution and aspect ratio adjustments compared to fixed square configurations.

**Scaling of Image Resolution.** We scale up image resolution with fixed square sizes for convenience (448 x 448 $\rightarrow$ 896 x 896$\rightarrow$ 1,344 x 1,344).

As shown in Table C.2, larger image resolution generally improves the model performance, particularly on web and desktop UIs that often contain small links and icons. However, mobile UIs, as being less dense, do not benefit as significantly from increased resolution.

**Dynamic Image Resolution and Aspect Ratio.** As shown in Table C.2, UGround benefits from dynamic image resolution supported by `AnyRes`, effectively adapting to varied resolutions and aspect ratios (for example, to mobile UIs or desktop UIs). This flexibility results in improved performance across platforms. For example, on desktop and web UIs, UGround achieves comparable or superior results using approximately 2/3 of the tokens required by the fixed 1,344 x 1,344 model in 16:9 scenarios.

Similar findings around these two aspects are also discussed in general domains (Li et al., 2024a; Zhang et al., 2024b), as well as some concurrent GUI works (Chen et al., 2024; Li et al., 2024c).

## C.3   RE TYPES

The taxonomy for REs introduced in this work represents a novel contribution and has not been addressed in prior studies (Li et al., 2020b; Hong et al., 2024; Cheng et al., 2024). In this section, we present ablation studies focused on the role of positional REs. We omit detailed studies on

---

[7]The data is converted to the format used in SeeClick. Given the maximum sequence length used in the training of Qwen-VL and SeeClick, we reduce the elements to a maximum of 30 for each page.

Table C.3: RE ablations for UGround on ScreenSpot (Agent Setting).

| Training Data | Mobile | | Desktop | | Web | | Average |
|---|---|---|---|---|---|---|---|
| | Text | Icon/Widget | Text | Icon/Widget | Text | Icon/Widget | |
| Web-Hybrid (w/o Pos REs) | 86.5 | 73.4 | 87.1 | 61.4 | 82.2 | **65.5** | 76.0 |
| Web-Hybrid | **89.0** | 73.4 | **88.1** | 61.4 | **84.8** | 64.6 | **76.9** |

Table C.4: RE ablations for UGround on ScreenSpot (Standard Setting).

| Training Data | Mobile | | Desktop | | Web | | Average |
|---|---|---|---|---|---|---|---|
| | Text | Icon/Widget | Text | Icon/Widget | Text | Icon/Widget | |
| Web-Hybrid (w/o Pos REs) | 72.2 | 52.0 | 72.7 | **55.0** | 76.5 | 61.2 | 64.9 |
| Web-Hybrid | **75.5** | **54.2** | **79.9** | **58.6** | **77.0** | **68.0** | **68.8** |

visual and functional REs because (1) they are interleaved in HTML DOMs and are challenging to fully disentangle, and (2) they have been extensively studied in prior work. For example, an HTML attribute (e.g., `aria-label`) may convey both visual and functional cues, and the MLLM can exploit different aspects of the input.

We train a new checkpoint with Web-Hybrid, omitting all positional REs while maintaining the overall number of web elements. As shown in Table C.3 and Table C.4, the inclusion of positional REs generally enhances model performance.

We hypothesize that the integration of positional and contextual data enables the model to more effectively capture and attend to the spatial relationships among UI elements. This enhanced contextual understanding is crucial for grounding tasks that cannot rely solely on visual or functional cues, especially in challenging cases where those cues alone are insufficient.

# D  EXAMPLES

## D.1  MULTIMODAL-MIND2WEB

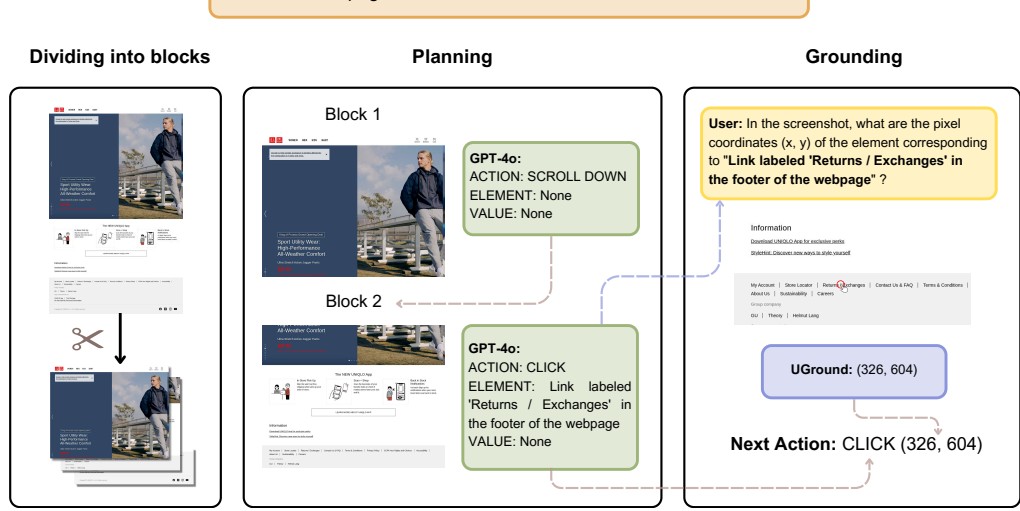

Figure D.1: Example of the Multimodal-Mind2Web evaluation pipeline.

## D.2 ANDROIDCONTROL

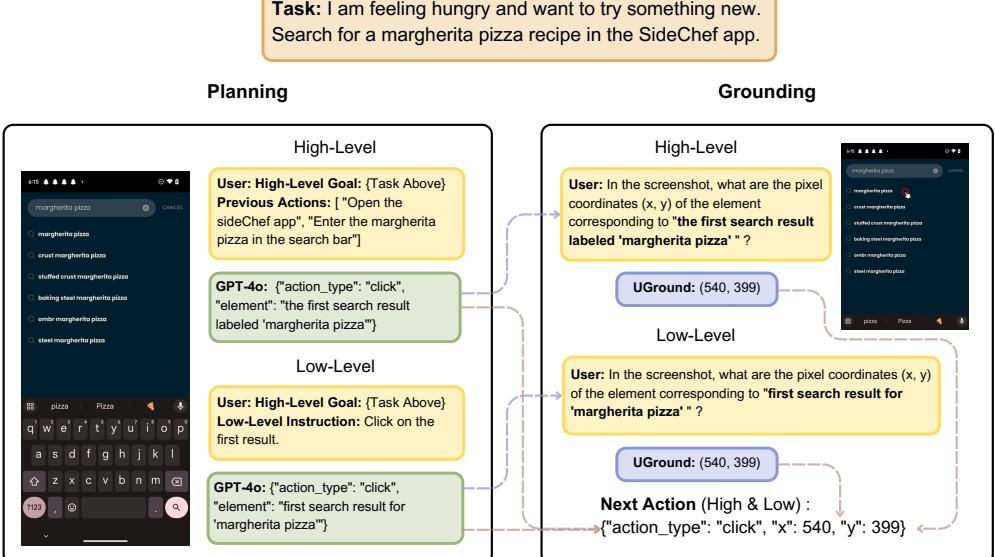

Figure D.2: Example of the AndroidControl evaluation pipeline.

## D.3 OMNIACT

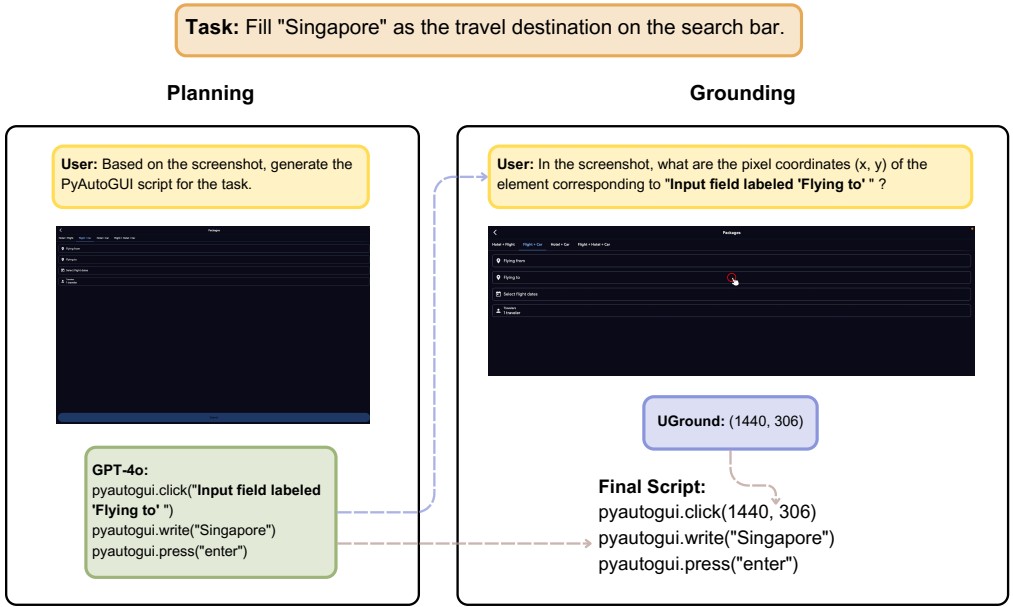

Figure D.3: Example of the OmniACT evaluation pipeline.

## D.4 TRAINING DATA

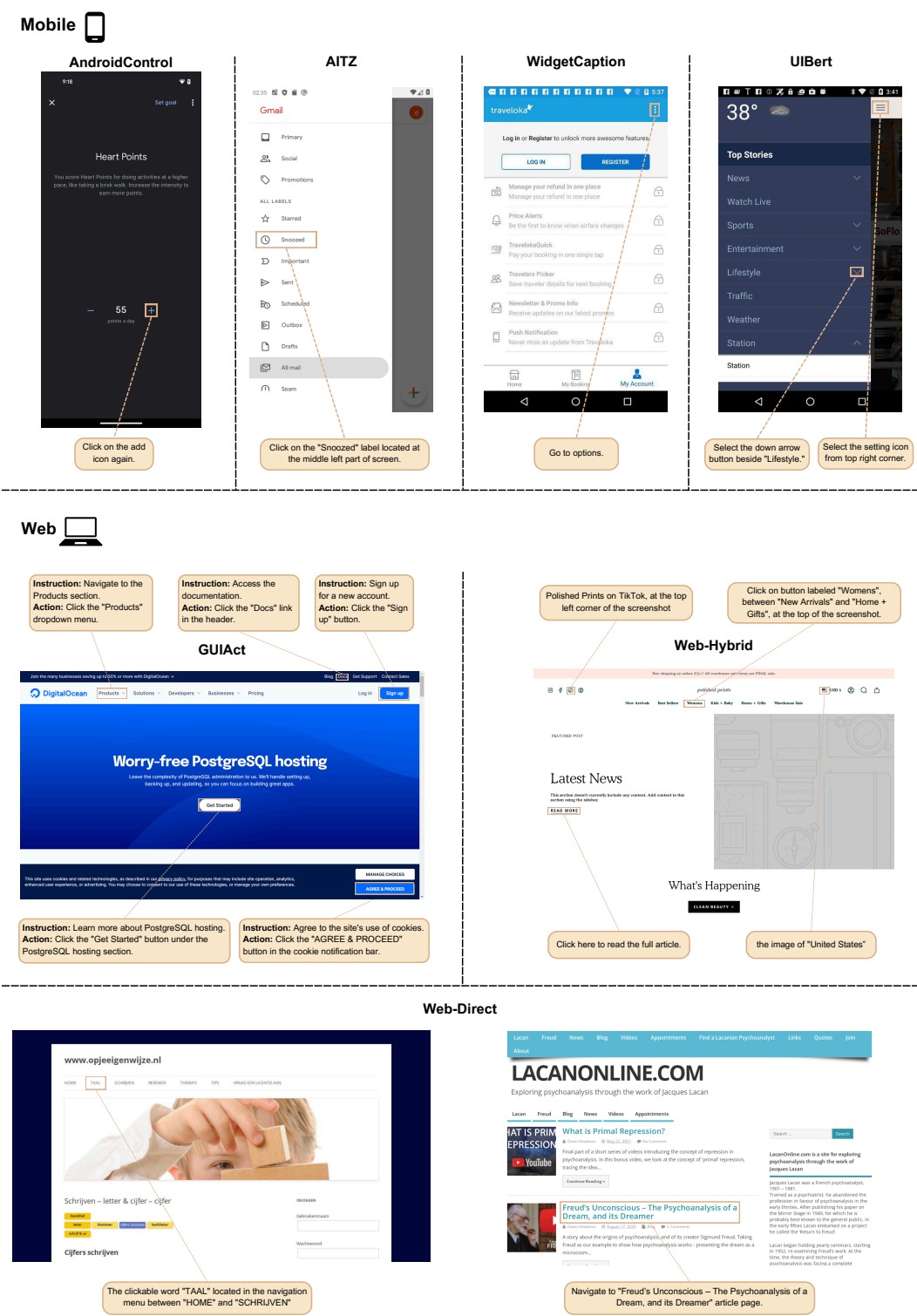

Figure D.4: Examples of training data from different sources.

# E   DATA CONSTRUCTION

We describe the details of our data construction in this section. Illustrative examples of all our training data are provided in Figure D.4.

## E.1   WEB-HYBRID

Following prior work (Hong et al., 2024; Cheng et al., 2024), we download and randomly sample from the latest Common Crawl[8]. We apply several filtering methods to exclude non-webpage files based on URL patterns and to remove non-English pages as indicated by the language labels provided by Common Crawl. We employ Playwright to load and render webpages, capture screenshots, and collect metadata for web elements. To ensure a diverse set of data, we simulate vertical scrolling to capture screenshots and elements at various positions on each webpage. The metadata for each element includes bounding box coordinates and relevant HTML attributes, such as the element's tag, inner text (inner_text), and alternative text (e.g., alt).

During rendering, we randomly select image sizes to cover a diverse range of resolutions and aspect ratios. Approximately one-third of the data is rendered in mobile-friendly aspect ratios, thereby triggering the mobile version of certain websites and enhancing the coverage of mobile UI environments. For each long webpage, up to three blocks of content within a viewport-sized area are randomly sampled to ensure content diversity. In total, the dataset comprises approximately 773K screenshots from around 700K URLs.

As detailed in §2.2, we employ a hybrid strategy to generate REs for webpage elements. Below, we first describe how we leverage MLLMs (LLaVA-NeXT-13B) and LLMs (Llama-3-8B) to generate concise, element-level descriptions without positional or contextual information.

We extract the bounding box regions from the webpage screenshots corresponding to the elements and pass these smaller cropped element images along with their salient HTML attributes to LLaVA. Using the prompts outlined below, we prompt LLaVA to generate an element description based on its internal knowledge, the element's image, and relevant HTML attributes:

> Based on the attached image of a web element, please provide a short description of the web element displayed. The goal is to capture the intuitive and visual appearance of the element. Use the accompanying HTML information as context but focus more on describing what is visually observable. Avoid directly referencing HTML attributes; instead, interpret their possible visual implications if they can be inferred from the image. Be cautious of potential inaccuracies in the HTML attributes and use them to enhance understanding only when they align reasonably with what can be inferred visually.
>
> HTML: {A list of salient HTML attributes}

We observe that since the input to LLaVA is a small cropped image, the model tends to have less hallucinations compared to directly caption an element with a bounding box overlaid in the image. However, due to the limited language capabilities of the 13B LLaVA model, the generated interpretations tend to be lengthy. To address this, the lengthy output is subsequently processed by Llama-3-8B with the prompt below that instructs it to condense the description into a brief referring expression:

> Here is a description of an element in a webpage. Using the detailed description provided, create a concise phrase that captures the essential visual and functional characteristics of the web element. The rephrased description should be straightforward, simple and precise enough to allow humans quickly spot this element in a webpage screenshot. Focus on the most prominent visual features and any critical function indicated by the text.
>
> Description: {}
>
> Leave only your final description in the answer, without any explanation.

Next, the generation process for each crawled element is as follows.

We begin by categorizing the webpage elements based on their tags into two groups: interactive elements (e.g., a, input, select, etc.) and pure text elements (e.g., p, h1, h2, etc.). Referring expressions are generated only

---

[8]CC-MAIN-2023-50

Table E.1: Statistics of element types (by HTML tags) in Web-Hybrid (%).

| a | img | button | input | svg | select | textarea | video |
|---|-----|--------|-------|-----|--------|----------|-------|
| 68.99 | 15.41 | 6.81 | 5.32 | 2.25 | 0.99 | 0.18 | 0.04 |

Table E.2: Statistics of element HTML attributes and MLLM-based synthetic REs used in Web-Hybrid (%). Calculated as the number of elements using an attribute/RE divided by the total number of elements.

| MLLM-based RE | inner-text | title | alt | aria-label | aria-describedby | placeholder | value |
|---------------|------------|-------|-----|------------|------------------|-------------|-------|
| 11.19 | 43.58 | 20.01 | 12.25 | 11.32 | 0.21 | 0.06 | 0.02 |

for interactive elements, as these constitute the primary targets in GUI grounding tasks. In addition, pure text elements are utilized as potential sources for referring expression generation.

For each interactive element, we first apply an OCR model (EasyOCR[9]) to extract text from the element's bounding box. If the similarity between the OCR-extracted text and the element's inner_text exceeds a threshold of 0.7, the element is considered textual, and the MLLM-based synthesis pipeline is bypassed. This procedure prevents the generation of trivial data (e.g., "Gray links labeled by link text"). Moreover, for textual elements, those sharing identical text with other elements on the same page are filtered out to avoid grounding ambiguities.

Based on manually crafted rules, we label each element's neighboring elements in various directions (multiple neighbors are allowed), mark the nearest upper h1, h2, or h3 elements (titles), and determine its absolute position (e.g., center of the screenshot, top, top-left corner) to generate position-based referring expressions. We randomly select up to neighboring elements in different directions and randomly pick elements whose distance from the target is within 500 pixels (empirically, always selecting the closest element does not yield the best performance). These are used to generate relative position descriptions. Some of the relative descriptions are further randomly modified to common terms such as "next to" or "between". For contextual references, if an element is identified as a checkbox or radio button based on its HTML properties, it is assumed to have an associated label (e.g., "radio button for Yes"). If such labels are provided in the HTML attributes, they are used directly; otherwise, the nearest element on the same row (or column, if necessary) is selected as the label. Similar procedures are followed for input fields and select boxes. Additional expressions such as "under," "in," or "under section A" are generated based on the hierarchical structure of titles (primarily h1, h2, and h3). Attributes like title, alt, or aria-label are always considered as potential descriptors, typically contributing functional information.

Finally, for each element, descriptors from accessibility labels, the element's own text, or MLLM-based descriptions are randomly combined with absolute positional information (included on a random basis) and supplemented by between zero and two relative or contextual descriptions. For interactive elements such as radio buttons, the label is always included. In each webpage, up to 100 elements are selected, prioritizing those with accessibility labels or MLLM annotations. The number of pure text elements is limited to no more than three times the sum of elements with accessibility labels and those annotated via MLLMs (with a minimum of 10, or the total available elements, whichever is lower) to reduce the number of pure text elements. Additionally, unique accessibility labels and their frequencies are counted; labels occurring more than 1,000 times are downsampled to a maximum of 1,000 occurrences. For example, the label "Next" appears 13K times, and is downsampled to 1K occurrences in our training data.

To illustrate the primary data distribution, we provide statistics about HTML element types, as well as attributes and positional RE types used in the final REs within Web-Hybrid. The statistics are shown in Table E.1, Table E.2, and Table E.3. We omit exact percentages of visual and functional REs because they are often interleaved in HTML DOMs and MLLM-based synthetic REs, and generally are hard to distinguish.

## E.2 WEB-DIRECT

For the Web-Direct dataset, we directly employ GPT-4o to generate referring expressions. We observed that, due to its limited grounded understanding capabilities, simply enclosing an element in the image with a bounding box often leads to notable hallucinations, particularly when it provides descriptions of nearby elements. To mitigate these hallucinations without incurring the high cost of manual post-verification, we find that annotating an element with both a red bounding box and a red arrow pointing to it substantially reduces hallucinations.

---

[9]https://github.com/JaidedAI/EasyOCR/

Table E.3: Statistics of relative positional REs, absolute Positional REs, and contextual REs used in Web-Hybrid (%). Contextual References are also counted as relative positional REs. Calculated as the number of elements using an RE divided by the total number of elements.

| Relative Positional RE | Contextual RE | Absolute Positional RE |
|:---:|:---:|:---:|
| 23.49 | 8.43 | 3.05 |

In addition, we explicitly query GPT-4o regarding the identification of the element, which further minimizes potential hallucinations and filters out a small number of crawling errors or occluded elements.

Two separate prompts are used in Web-Direct: one to generate free-form referring expressions and another to generate functionally oriented referring expressions:

> Here is supposed to be an interactive element (button, link, dropdown, text box, etc.) in the red box pointed by an arrow in the screenshot. Can you find it? Is it visible from the screenshot? **Can you write a concise description that is sufficient for humans to locate it from the screenshot**? Your response should be a JSON. For example, "visible": true, "description": "your description here".

> Here is supposed to be an interactive element (button, link, dropdown, text box, etc.) in the red box pointed by an arrow in the screenshot. Can you find it? Is it visible from the screenshot? **What unique function does this element enable**? Your response should be a JSON. For example, "visible": true, "action": "subscribe the latest updates".

### E.3 OPEN-SOURCE DATA

We leverage several high-quality open-source referring expression datasets in Android, as well as the GUIAct dataset, as supplementary sources of web data. Specifically:

1. **GUIAct**: We use the annotated data from GUIAct (web-single). Steps that do not involve coordinates or that are marked as multi-step operations (for example, "click ... then type") are filtered out. We use both the *Instruction* and *Action* annotations for grounding (i.e., each element is seen in training twice with different expressions).

2. **AndroidControl**: Similarly, we use the human-annotated actions from the training set. We filter out any actions that do not have associated coordinate data, ensuring that only steps with specific visual grounding targets are included in the dataset.

3. **Widget Caption**: For each element in the training set, multiple functional captions are provided. To enhance diversity, two captions per element are randomly selected from the available set of functional captions during data construction.

4. **UIBert**: We use the training set elements from UIBert without any additional special processing, directly utilizing the referring expressions provided by this dataset.

5. **AITZ**: We incorporate the annotated actions (`Thought`) from AITZ, using each step's action annotation for grounding in the dataset. These annotations contribute to a more diverse set of referring expressions, particularly for action-oriented grounding tasks.

## F MODEL AND TRAINING DETAILS

### F.1 OVERVIEW

For flexible investigation of the model architecture, we build the architecture based on LLaVA-NeXT (Liu et al., 2024b), and train from scratch using open-source data from Liu et al. (2024a). We use CLIP-ViT-L-14 (224px) as our base image encoder for more flexible splitting of `AnyRes`, and keep it frozen during training. We use Vicuna-1.5-7b-16k (Zheng et al., 2023) as the language backbone as a long-context LM backbone for handling long visual contexts.

## F.2 ANYRES

As described in §2.3, `AnyRes` allows convenient scaling up of image resolutions, although it's not always beneficial to enlarge image resolutions (Li et al., 2024a). We keep the main pipeline of `AnyRes`, splitting images into 224px grids. However, to keep the original image aspect ratios, we resize only by width and pad to the bottoms if needed, and use pixel-level coordinates in numbers that are compatible with this design. We allow at most 36 grids, for a maximum resolution of 1,344 x 1,344 and 896 x 2,016. We empirically find `AnyRes` does not generalize to unseen image resolutions for visual grounding. Therefore, we resize images by width to keep them within the training resolution ranges when needed. We remove the low-resolution image for providing global context, because it intuitively does not provide informative contexts when images are larger than 1,000px, and we empirically find it slightly hurt the performance.

## F.3 TRAINING

Our training primarily consists of two stages:

1. **LLaVA-1.5 Pretraining and Finetuning**: We follow the exact pretraining in Liu et al. (2024a). Then, in the instruction finetuning stage, we change the grounding data from normalized coordinates to absolute coordinates as we wish, and start to use our modified `AnyRes` setting.

2. **GUI Visual Grounding**: Then we train UGround on our training datasets.

Due to the huge computation cost of handling high-resolution images, we use LoRA (Hu et al., 2022) for instruction finetuning in the two stages, with a device batch size of 4.

The first stage takes about 50 hours on a single 4x NVIDIA A100 machine (global batch size 128 with gradient accumulation). For the large-scale GUI data training, we use 112 NVIDIA H100 GPUs and finish the training in about 6 hours (global batch size 448).

# G   EVALUATION DETAILS

## G.1   MODEL ENDPOINTS

As studied in (Pan et al., 2024), different GPT endpoints could lead to slight differences in the performance of GUI tasks. Hence, we provide the specific endpoint names we use in our evaluation, as well as those of the baselines we use (if available).

- Ours (across every benchmark): `gpt-4-turbo-2024-04-09` and `gpt-4o-2024-05-13`
- Multimodal-Mind2Web: `gpt-4-1106-vision-preview`
- OmniACT: `gpt-4-0613` and `gpt-4-1106-vision-preview`
- Mind2Web-Live: `gpt-4-0125-preview` and `gpt-4o-2024-05-13`
- AndroidWorld: `gpt-4-turbo-2024-04-09`

## G.2   MULTIMODAL-MIND2WEB

Many screenshots in Multimodal-Mind2Web have giant vertical heights (e.g., $1,280 \times 10,000$ pixels). Similar to Zheng et al. (2024), to avoid overly long screenshots, we divide whole webpage screenshots into viewport-sized blocks, and simulate scrolling down to the next block whenever agents determine that no valid action can be taken or explicitly choose to scroll. Specifically, we divide each full-page screenshot into $1,280 \times 1,000$ pixel blocks, except for the final block, which may be shorter depending on the page's total height. Most of the target elements are within the first block (about 80%). See Figure D.1 for an illustrative example of the pipeline.

We report *element accuracy* on the benchmark, and the grounding is considered to be correct if the output coordinates fall in the box coordinates of the ground truth element.

## G.3   ANDROIDCONTROL

We adopt the M3A (Multimodal Autonomous Agent for Android) prompt (Rawles et al., 2024), the state-of-the-art zero-shot method in Li et al. (2024b). We only make minor modifications to integrate UGround into M3A.

We follow the standard data processing steps outlined in Li et al. (2024b). During evaluation, coordinates generated by grounding models are translated to the smallest visible element that includes the coordinates.

### G.4 OMNIACT

We follow the method in Kapoor et al. (2024) for prompt design and the selection of five in-context examples. The prompt is slightly modified to generate element descriptions as function parameters for PyAutoGUI scripts, instead of directly outputting coordinates. After generating the PyAutoGUI script with element descriptions, we use grounding models to predict the corresponding coordinates and substitute them back into the original script. See Figure D.3 for an illustrative example of the pipeline.

We compare our method with DetACT (Kapoor et al., 2024), the state-of-the-art method in Kapoor et al. (2024), which extracts UI elements and their coordinates through a combination of OCR, icon matching, and color detection. These elements are filtered by task relevance and passed to LLMs or MLLMs to generate the PyAutoGUI script. In contrast, our method does not use a pre-generated elements list. The planner model focuses on generating precise element descriptions based solely on the screenshot. Additionally, we corrected basic errors in the public evaluation scripts (for example, wrong file paths and wrong calculation of distances).

### G.5 MIND2WEB-LIVE

The baseline agent in Pan et al. (2024) is text-only, perceives and interacts with webpages by hundreds of textual HTML elements at a time. To study vision-only agents, we change the observation to pure screenshots. We also make necessary changes to the standard action space to entirely isolate HTML from the planning, grounding, and execution: 1) We add `Scroll_Up` and `Scroll_Down` to the action space to better support vision-only agents with viewport-sized observation. 2) We remove `Fill_Form` and `Fill_Search` from the action space, which use an additional judgment model to determine whether to press enter after typing through HTML information. Instead, we use `Type` and `Press_Enter` to let the agent make its own decisions autonomously. 3) We disable API-based `Select`, and force agents to select options merely through clicking and make the action more challenging. We admit some select buttons cannot be easily operated with only `Click`. We compromise this point to fulfill the motivation of this vision-only study.

### G.6 ANDROIDWORLD

We build SeeAct-V agents based on the M3A agent in Rawles et al. (2024), which receives both raw and SoM images, and reason about the next action in a ReAct style (Yao et al., 2023) and choose the next target element from the element list. It also adopts self-reflection (Shinn et al., 2024) in the agent pipeline to instruct agents to summarize the current move and facilitate the following steps.

We mainly remove SoM images and textual list of elements from the a11y tree in the observation (in both planning and reflection phases), and change element-based actions to pixel-level actions.

# H PROMPTS

Table H.1: Prompt used for the planning model in **Multimodal-Mind2Web**, modified from the prompt in (Zheng et al., 2024)

---

**System Role**
You are imitating humans doing web navigation for a task step by step.
At each stage, you can see the webpage like humans by a screenshot and know the previous actions before the current step through recorded history.
You need to decide on the first following action to take.
You can click an element with the mouse, select an option, type text with the keyboard, or scroll down.

---

**Task Description**
You are asked to complete the following task: {Task description}
Previous Actions: {List of previous actions, if any}
The screenshot below shows the webpage you see.

---

**Useful Guidelines**
First, observe the current webpage and think through your next step based on the task and previous actions.

To be successful, it is important to follow the following rules:
1. Make sure you understand the task goal to avoid wrong actions.
2. Ensure you carefully examine the current screenshot and issue a valid action based on the observation.
3. You should only issue one action at a time.
4. The element you want to operate with must be fully visible in the screenshot. If it is only partially visible, you need to SCROLL DOWN to see the entire element.
5. The necessary element to achieve the task goal may be located further down the page. If you don't want to interact with any elements, simply select SCROLL DOWN to move to the section below.

---

**Reasoning**
Explain the action you want to perform and the element you want to operate with (if applicable).
Describe your thought process and reason in 3 sentences.

---

**Output Format**
Finally, conclude your answer using the format below.
Ensure your answer strictly follows the format and requirements provided below, and is clear and precise.
The action, element, and value should each be on three separate lines.

ACTION: Choose an action from CLICK, TYPE, SELECT, SCROLL DOWN. You must choose one of these four, instead of choosing None.

ELEMENT: Provide a description of the element you want to operate. (If ACTION == SCROLL DOWN, this field should be none.)
It should include the element's identity, type (button, input field, dropdown menu, tab, etc.), and text on it (if applicable).
Ensure your description is both concise and complete, covering all the necessary information and less than 30 words.
If you find identical elements, specify its location and details to differentiate it from others.

VALUE: Provide additional input based on ACTION.
The VALUE means:
If ACTION == TYPE, specify the text to be typed.
If ACTION == SELECT, specify the option to be chosen.
Otherwise, write 'None'.

---

Table H.2: Prompts used for the planning model in **AndroidControl**, modified from the prompt in (Li et al., 2024b) and (Rawles et al., 2024)

---

**General Instruction**
You are an agent who can operate an Android phone on behalf of a user.
Based on user's goal/request, you may complete some tasks described in the requests/goals by performing actions (step by step) on the phone.

When given a user request, you will try to complete it step by step. At each step, you will be given the current screenshot and a history of what you have done (in text). Based on these pieces of information and the goal, you must choose to perform one of the action in the following list (action description followed by the JSON format) by outputting the action in the correct JSON format.
- If you think the task has been completed, finish the task by using the status action with complete as goal_status: {"action_type":"status","goal_status":"successful"}
- If you think the task is not feasible (including cases like you don't have enough information or cannot perform some necessary actions), finish by using the 'status'action with infeasible as goal_status: {"action_type": "status", "goal_status": "infeasible"}
- Click/tap on an element on the screen, describe the element you want to operate with: {"action_type": "click", "element": ⟨target_element_description⟩}
- Long press on an element on the screen, similar with the click action above: {"action_type": "long_press", "description": ⟨target_element_description⟩}
- Type text into a text field: {"action_type": "type_text", "text": ⟨text_input⟩, "element": ⟨target_element_description⟩}
- Scroll the screen in one of the four directions: {"action_type": "scroll", "direction": ⟨up, down, left, right⟩}
- Navigate to the home screen: {"action_type": "navigate_home"}
- Navigate back: {"action_type": "navigate_back"}
- Open an app (nothing will happen if the app is not installed): {"action_type": "open_app", "app_name": ⟨name⟩}
- Wait for the screen to update: {"action_type": "wait"}

---

**Useful Guidelines**
Here are some useful guidelines you need to follow:
General:
- Usually there will be multiple ways to complete a task, pick the easiest one. Also when something does not work as expected (due to various reasons), sometimes a simple retry can solve the problem, but if it doesn't (you can see that from the history), SWITCH to other solutions.
- If the desired state is already achieved (e.g., enabling Wi-Fi when it's already on), you can just complete the task.

Action Related:
- Use the 'open_app' action whenever you want to open an app (nothing will happen if the app is not installed), do not use the app drawer to open an app unless all other ways have failed.
- Use the 'type_text' action whenever you want to type something (including password) instead of clicking characters on the keyboard one by one. Sometimes there is some default text in the text field you want to type in, remember to delete them before typing.
- For 'click', 'long_press' and 'type_text', the element you pick must be VISIBLE in the screenshot to interact with it.
- The 'element' field requires a concise yet comprehensive description of the target element in a single sentence, not exceeding 30 words. Include all essential information to uniquely identify the element. If you find identical elements, specify their location and details to differentiate them from others.
- Consider exploring the screen by using the 'scroll' action with different directions to reveal additional content.
- The direction parameter for the 'scroll' action specifies the direction in which the content moves and opposites to swipe; for example, to view content at the bottom, the 'scroll' direction should be set to 'down'.

Text Related Operations:

---

*Continued on the next page*

Table H.2 – Continued from the previous page

---

- Normally to select certain text on the screen: ⟨i⟩ Enter text selection mode by long pressing the area where the text is, then some of the words near the long press point will be selected (highlighted with two pointers indicating the range) and usually a text selection bar will also appear with options like 'copy', 'paste', 'select all', etc. ⟨ii⟩ Select the exact text you need. Usually the text selected from the previous step is NOT the one you want, you need to adjust the range by dragging the two pointers. If you want to select all text in the text field, simply click the 'select all' button in the bar.
- At this point, you don't have the ability to drag something around the screen, so in general you cannot select arbitrary text.
- To delete some text: the most traditional way is to place the cursor at the right place and use the backspace button in the keyboard to delete the characters one by one (can long press the backspace to accelerate if there are many to delete). Another approach is to first select the text you want to delete, then click the backspace button in the keyboard.
- To copy some text: first select the exact text you want to copy, which usually also brings up the text selection bar, then click the 'copy' button in bar.
- To paste text into a text box, first long press the text box, then usually the text selection bar will appear with a 'paste' button in it.
- When typing into a text field, sometimes an auto-complete dropdown list will appear. This usually indicates this is a enum field and you should try to select the best match by clicking the corresponding one in the list.

---

**High-Level Prompt**
{General Instruction}
The current user goal/request is: {High-level goal}
Here is a history of what you have done so far: {History}

The current raw screenshot is given to you.
{Useful Guidelines}

Now output an action from the above list in the correct JSON format, following the reason why you do that. Your answer should look like:
Reason: ...
Action: {"action_type": ...}

Your Answer:

---

**Low-Level Prompt**
{General Instruction}
The user's high-level goal/request is: {High-level goal}
The current next step's low-level goal is: {Low-level goal}

The current raw screenshot is given to you.
{Useful Guidelines}

Now output an action from the above list in the correct JSON format, following the reason why you do that. Your answer should look like:
Reason: ...
Action: {"action_type": ...}

Your Answer:

---

Table H.3: Prompt used for the planning model in **OmniACT**, modified from the prompt in (Kapoor et al., 2024)

---

**General Instruction**
You are an excellent robotic process automation agent who needs to generate a PyAutoGUI script for the tasks given to you.
You will receive some examples to help with the format of the script that needs to be generated.

There are some actions that require you to provide an element description for the elements you want to operate on. For the description, follow the requirements below:
Element Description Requirements:
Provide a concise description of the element you want to operate.
It should include the element's identity, type (button, input field, dropdown menu, tab, etc.), and text on it (if have).
If you find identical elements, specify their location and details to differentiate them from others.
Ensure your description is both concise and complete, covering all the necessary information and less than 30 words, and organize it into one sentence.

[IMPORTANT!!] Stick to the format of the output scripts in the example.
[IMPORTANT!!] Use only the functions from the API docs.
[IMPORTANT!!] Follow the output format strictly. Only write the script and nothing else.

---

**API Reference**
Here is the API reference for generating the script:
def click(element=description):
'''Moves the mouse to the element corresponding to the description and performs a left click.
Example:
High Level Goal: Click at the rectangular red button labeled "Next".
Python script:
import pyautogui
pyautogui.click("Rectangular red button labeled "Next" ")
'''
pass

def rightClick(element=description):
'''Moves the mouse to the element corresponding to the description and performs a right click.
Example:
High Level Goal: Right-click at link labeled "vacation rentals"under the "housing"section.
Python script:
import pyautogui
pyautogui.rightClick("Link labeled "vacation rentals"under the "housing"section")
'''
pass

def doubleClick(element=description):
'''Moves the mouse to the element corresponding to the description and performs a double click.
Example:
High Level Goal: Double-click at folder named "courses".
Python script:
import pyautogui
pyautogui.doubleClick("Folder named "courses" ")
'''
pass

def scroll(clicks=amount_to_scroll):
'''Scrolls the window that has the mouse pointer by float value (amount_to_scroll).
Example:
High Level Goal: Scroll screen by 30.
Python script:
import pyautogui
pyautogui.scroll(30)
'''
pass

---

Table H.3 – Continued from the previous page

```
def hscroll(clicks=amount_to_scroll):
'''Scrolls the window that has the mouse pointer horizontally by float value (amount to scroll).
Example:
High Level Goal: Scroll screen horizontally by 30.
Python script:
import pyautogui
pyautogui.hscroll(30)
'''
pass

def dragTo(element=description, button=holdButton):
'''Drags the mouse to the element corresponding to the description with (holdButton) pressed. hold-
Button can be 'left', 'middle', or 'right'.
Example:
High Level Goal: Drag the screen from the current position to recycle bin with the left click of the
mouse.
Python script:
import pyautogui
pyautogui.dragTo("Recycle bin with trash can shape", "left")
'''
pass

def moveTo(element = description):
'''Takes the mouse pointer to the element corresponding to the description.
Example:
High Level Goal: Hover the mouse pointer to search button.
Python script:
import pyautogui
pyautogui.moveTo("Request appointment button")
'''
pass

def write(str=stringType, interval=secs_between_keys):
'''Writes the string wherever the keyboard cursor is at the function calling time with
(secs_between_keys) seconds between characters.
Example:
High Level Goal: Write "Hello world"with 0.1 seconds rate.
Python script:
import pyautogui
pyautogui.write("Hello world", 0.1)
'''
pass

def press(str=string_to_type):
'''Simulates pressing a key down and then releasing it up. Sample keys include 'enter', 'shift', arrow
keys, 'f1'.
Example:
High Level Goal: Press the enter key now.
Python script:
import pyautogui
pyautogui.press("enter")
'''
pass

def hotkey(*args = list_of_hotkey):
'''Keyboard hotkeys like Ctrl-S or Ctrl-Shift-1 can be done by passing a list of key names to hotkey().
Multiple keys can be pressed together with a hotkey.
Example:
High Level Goal: Use Ctrl and V to paste from clipboard.
Python script:
import pyautogui
```

Table H.3 – Continued from the previous page

| |
| --- |
| pyautogui.hotkey("ctrl", "v")
"""
pass |

| **Examples** |
| --- |
| Here are some examples similar to the tasks you need to complete.
However, these examples use coordinate format for actions like click, rightClick, doubleClick, moveTo, dragTo, instead of element description.
You should only refer to the actions in these examples, and for the output format, stick to the content in the API reference.
For example, do not output "pyautogui.click(100,200)", instead output "pyautogui.click("Gray Tools menu button with a downward arrow in the top right corner") ".
Omit "import pyautogui", do not include any comments or thoughts. Your output should only contain the script itself.
{Example list} |

| **Task Description** |
| --- |
| Based on the screenshot, generate the PyAutoGUI script for the following task: {Task description}
You should list all the necessary steps to finish the task, which could involve multiple steps. Also, ensure simplifying your steps as much as possible, avoid dividing a single task into multiple steps if it can be completed in one. |

Table H.4: Prompt used for the planning model in **ScreenSpot (Agent Setting)**.

| **Task Description** |
| --- |
| You are an excellent agent for mobile, web, and desktop navigation tasks.
Describe the target element for this task based on the provided screenshot:
Task: {Task description} |

| **Element Description Requirements** |
| --- |
| Provide a concise description of the element you want to operate.
Ensure your description is both concise and complete, covering all the necessary information in less than 30 words, and organized into one sentence.
If you find identical elements, specify their location and details to differentiate them from others. |

| **Output Format** |
| --- |
| Your output should only include the element description itself and follow the requirements.
Do not start with "the target element" or "the element". |

