# OpenReview forum: "Navigating the Digital World as Humans Do: Universal Visual Grounding for GUI Agents"
_ICLR.cc/2025/Conference — ICLR 2025 Oral_

### Official Review · Reviewer_Xm5m · 2024-10-19

**Soundness:** 3
**Presentation:** 4
**Contribution:** 3
**Rating:** 10
**Confidence:** 5

**Summary:**

The paper presents a novel approach to enhancing GUI agents by adopting a human-like embodiment that solely relies on visual perception and various operations. It addresses the limitations of current GUI agents that depend on text-based representations (HTML, accessibility trees, etc.), which often introduce noise and computational overhead. The authors propose UGround, a visual grounding model trained on a large dataset of GUI elements (19M GUI elements across 1.3M screenshots). Their framework, SeeAct-V, allows GUI agents to perceive and interact with their environment visually, improving grounding accuracy and agent performance. The empirical evaluation across six benchmarks shows that UGround outperforms state-of-the-art models, offering a promising solution for GUI agents to function more like humans in digital environments.

**Strengths:**

I think this paper addresses well on the bottleneck of MLLM-based GUI Agents, UI Grounding, with a newly constructed synthetic dataset, which is timely and important.

- **Human-Like Embodiment for GUI Agents**: The paper makes a compelling argument for GUI agents that perceive their environment visually, which aligns better with how humans interact with GUIs.
- **Extensive Grounding Dataset**: UGround is trained on a comprehensive dataset of 19 million GUI elements, making it the largest dataset for GUI visual grounding.
- **Significant Performance Gains**: The empirical results demonstrate that UGround improves grounding accuracy by up to 20%, and agents using it outperform models that rely on both visual and text-based inputs.
- **Cross-Platform Generalization**: UGround shows strong performance across different platforms (desktop, web, mobile), which demonstrates its potential as a universal solution.
- Comprehensive Evaluation: The authors evaluate UGround on six benchmarks, which span grounding tasks and online/offline agent evaluations, providing a thorough analysis of the model's capabilities.

**Weaknesses:**

- **LLM Usage in Synthesizing Data**: While the dataset is large, much of it is synthetically generated using LLMs. This raises concerns about potential hallucinations during data synthesis. The authors should consider sampling a subset of the data for human evaluation to verify the accuracy and goal alignment of the generated content, especially when leveraging models like LLaVA-Next-13B and GPT-4o for generating and refining referring expressions (REs), as well as LLaMA-3-8B in polishing. Here is a potential experiment setup suggested by GPT's feedback: `Randomly sample 1000 generated referring expressions and have human annotators rate their accuracy and relevance.` I think this advice is acceptable.
- **Dataset Analysis**: A deeper analysis of the dataset's diversity, as mentioned in line 169, would strengthen the work. Techniques such as PCA could provide insights into the data distribution. Additionally, more information is needed on how well the instructions used for planning in GUI agents align with the dataset used for training the grounding model. A breakdown of the types of GUI elements represented in the dataset, or t-SNE plots to visualize the distribution of referring expressions is suggested.
- **Typos**: There is a citation typo in line 307 related to CogAgent, which should be corrected.
- **Copyright Concerns**: The paper uses webpage data crawled from Common Crawl. It would be helpful for the authors to address any potential copyright issues associated with using this data. The authors are suggested to specifically discuss their data usage policy and any steps they've taken to ensure compliance with copyright laws when using Common Crawl data.
- **Environment Limitation**: The grounding dataset was collected entirely in a web environment, which represents only a subset of GUIs. The authors have discussed this limitation and I think it is not a big problem given that GUI elements can transfer smoothly. However, it's still worth emphasizing that this could impact the model’s performance in other GUI environments. Include a small-scale experiment or case study demonstrating the model's performance on a non-web GUI task is suggested.

**Questions:**

See Weakness above.

I think the authors can include a section to discuss copyright problems.

**Details Of Ethics Concerns:**

A discussion section of copyright problem is needed.

---

> ### Author Response · Authors · 2024-11-26
> **Part 1/2: Data Analysis; A Typo**
>
> > **(W1) LLM Usage in Synthesizing Data: While the dataset is large, much of it is synthetically generated using LLMs. This raises concerns about potential hallucinations during data synthesis. The authors should consider sampling a subset of the data for human evaluation to verify the accuracy and goal alignment of the generated content, especially when leveraging models like LLaVA-Next-13B and GPT-4o for generating and refining referring expressions (REs), as well as LLaMA-3-8B in polishing. Here is a potential experiment setup suggested by GPT's feedback: Randomly sample 1000 generated referring expressions and have human annotators rate their accuracy and relevance. I think this advice is acceptable.**
>
> Thank you for suggesting a more intrinsic evaluation of our data. We agree that would enhance our work.
>
> Firstly, a minor clarification:  our carefully curated rules are also crucial for improving the comprehensiveness of the referring expressions (REs), so it’s not just the LLMs.
>
> For the suggested intrinsic evaluation, we randomly sampled and analyzed 200 REs generated by the LLaVA-\>Llama pipeline, focusing on the hallucination and goal alignment of the final REs.
>
> We categorize the instances into the following:
>
> * **Acceptable:**
>   * **Correct:** Providing sufficient information for humans to unambiguously locate the element, with accurate details such as text and colors.
>   * **Slightly Hallucinated:** Containing minor inaccuracies but remaining sufficient for humans to locate the element.
>   * **Vague:** Overly general but still allowing humans to understand and locate the element.
> * **Unacceptable**:
>   * **Wrong:** Completely incorrect or too vague to understand.
>   * **Wrong Bbox:** Invalid due to incorrect element areas given to LLaVA rather than hallucinations (see examples below).
>
> **Categorization statistics:**
>
> * **Correct:** 122 / 200 \= 61.0%
> * **Slightly Hallucinated:** 37 / 200 \= 18.5%
>   * Main issues:
>     * Inaccurate colors
>     * Addition of non-existing text (mostly hallucinated from HTML context in the input)
>     * Inaccurate shapes, functions, or numbers
> * **Vague:** 5 / 200 \= 2.5%
>   * Main issues:
>     * Lack of specificity (e.g., "input field" without details).
>     * Incomplete text content.
> * **Wrong:** 7 / 200 \= 3.5%
>   * Main issues:
>     * Incorrect text.
>     * Misrepresented direction of arrows (⬅️➡️).
>     * Described as something else).
> * **Wrong Bbox:** 29/200= 14.5 %
>   * Main issues:
>     * Elements being blocked by other elements (e.g. pop-up ads)
>     * Elements moved during collecting due to some live changes in the page.
>
> Overall, **82% are deemed acceptable REs**. Another interesting and noteworthy point is that LLM fine-tuning, partially thanks to the strong priors from pre-training, seems to be **robust to a certain level of noise in the training data**. In a sense, LLMs seem to be able to learn the right behavior from the correct training data while not being distracted too much by the incorrect training data, likely because the incorrect data is usually far off of the prior LLMs have obtained from pre-training. This partly explains UGround’s strong empirical performance despite the noise in the synthesized data. But of course, this is mainly our speculation based on our observation. It’d be an interesting future work to quantitatively study this behavior.
>
> > **(W2) Dataset Analysis: A deeper analysis of the dataset's diversity, as mentioned in line 169, would strengthen the work. Techniques such as PCA could provide insights into the data distribution. Additionally, more information is needed on how well the instructions used for planning in GUI agents align with the dataset used for training the grounding model. A breakdown of the types of GUI elements represented in the dataset, or t-SNE plots to visualize the distribution of referring expressions is suggested.**
>
> Thanks for the suggestion. We agree that an in-depth data analysis will provide more insights into data quality and distribution, which will generally improve reproduction and further development. We will add a section in the Appendix to provide a more detailed breakdown of: 1\) **types of web elements** and 2\) **RE types** to help clarify the data distribution.
>
> > **(W3): Typos: There is a citation typo in line 307 related to CogAgent, which should be corrected.**
>
> Thanks for identifying this typo; we will correct it in the revised version.

---

> ### Author Response · Authors · 2024-11-26
> **Part 2/2: Copyright Concerns; Environment Limitation**
>
> > **(W4): Copyright Concerns**: The paper uses webpage data crawled from Common Crawl. It would be helpful for the authors to address any potential copyright issues associated with using this data. The authors are suggested to specifically discuss their data usage policy and any steps they've taken to ensure compliance with copyright laws when using Common Crawl data. **A discussion section of copyright problem is needed.**
>
> Thank you for raising these copyright concerns. **We will provide statements regarding our data usage in the revised version.** Here, we would like to discuss and clarify the copyright concerns.
>
>  **Data Collection Methodology:**
>
> Our data collection follows the prior works in the field of GUI visual grounding \[1, 2\] (our main baselines).
>
> \[1\] Hong, Wenyi, et al. CogAgent: A Visual Language Model for GUI Agents. CVPR 2024\.
>
> \[2\] Cheng, Kanzhi, et al. SeeClick: Harnessing GUI Grounding for Advanced Visual GUI Agents. ACL 2024\.
>
> **Data Accessibility and Licensing:**
>
> As the reviewer mentioned, the webpages we used are sourced from the Common Crawl dataset, which is a publicly available Internet archive designed for research and non-commercial use. We utilized only a small subset of it (773K webpages out of 3.35B) and strictly adhered to Common Crawl’s Terms of Use throughout our work.
>
>  **Legal Compliance:**
>
> Our use of this data is strictly for academic research purposes and is fully compliant with **Section 107 of the U.S. Copyright Law: *Limitations on exclusive rights: Fair use***. Specifically:
>
> 1. **Purpose and character of the use, including whether the use is of a commercial nature or is for nonprofit educational purposes:** Our work is solely for academic research purposes, with no intention of commercial use.
> 2. **Nature of the copyrighted work:** Our work does not involve the use of creative or imaginative works (e.g., novels, movies, or songs).
> 3. **Amount and substantiality of the portion used in relation to the copyrighted work as a whole:** We only use a small portion of the webpages, specifically rendered viewport-size screenshots and element coordinates. Our work does not utilize entire pages or their primary content. The trained models produce only coordinate outputs, not any copyrighted material.
> 4. **Effect of the use upon the potential market for or value of the copyrighted work:** Our use does not impact the commercial market or value of the original websites.
>
> We will include detailed explanations and statements regarding our data usage in the revised version, to ensure transparency and compliance with copyright regulations.
>
>
>
> > **W5: Environment Limitation: The grounding dataset was collected entirely in a web environment, which represents only a subset of GUIs. The authors have discussed this limitation and I think it is not a big problem given that GUI elements can transfer smoothly. However, it's still worth emphasizing that this could impact the model’s performance in other GUI environments. Include a small-scale experiment or case study demonstrating the model's performance on a non-web GUI task is suggested.**
>
> We agree with the point that web-only training could *potentially* limit the performance in other platforms. Tha said, as the reviewer has noticed, we do show strong **cross-platform generalization** of the dataset (from the ablation studies trained with only Web-Hyrbid in **Sec. 3.5** and in the response to **Reviewer bKGa**).
>
> To reiterate, we have comprehensively evaluated the model's performance in mobile and desktop environments:
>
> * **Mobile**:
>   * ScreenSpot-Mobile (Table 2, 3\)
>   * AndroidControl (Table 5\)
>   * AndroidWorld (Table 7\)
> * **Desktop**:
>   * ScreenSpot-Desktop (Table 2, 3\)
>   * OmniACT (Table 6\)
>
> From the results, we find that UGround works **reasonably well in non-web platforms**, not only substantially better than existing models on the grounding benchmark (ScreenSpot-Mobile and ScreenShot-Desktop) but also achieving SOTA results on mobile and desktop agent benchmarks (AndroidControl and OmniACT).
>
> We also provided error analysis in Sec. 3.4, showing that the **most of the errors in AndroidControl (90%+) are planning errors**, not grounding errors from UGround. We also barely observed grounding errors in the experiments in AndroidWorld.
>
> Nevertheless, in desktop environments, a considerable portion of the failures do stem from grounding. We have some discussion in Sec. 3.4, and we do have plans to further enhance grounding on Desktop UIs and long-tail elements in future work.

---

> ### Comment · Reviewer_Xm5m · 2024-12-03
> **Thank you for your rebuttal**
>
> Thank you for your detailed response and additional experiments. After reviewing your rebuttal, there are no concerns from my side. Furthermore, having examined the majority of GUI-related submissions to ICLR this year, I believe this paper stands out as one of the strongest contributions in the field. I suggest an oral award for this paper. Given the paper's exceptional quality, I am raising my score.

---

> > ### Author Response · Authors · 2024-12-03
> > **Thanks!**
> >
> > That's such a compliment! It really means a lot to us.

---

### Official Review · Reviewer_bKGa · 2024-10-30

**Soundness:** 2
**Presentation:** 3
**Contribution:** 2
**Rating:** 5
**Confidence:** 4

**Summary:**

This paper proposed a visual-based GUI Agent framework (SeeAct-V) with a GPT-aid planner and a visual grounding model (UGround). The planner first generates a textual plan, and the visual grounding model produces the coordinate of the target element. The paper constructs a training dataset (Web-Hybird) to train the grounding model. SeeAct-V is evaluated on 3 settings (visual grounding, offline and online agent evaluation).  The results validate UGround’s efficacy as a universal grounding model for GUI agents.

**Strengths:**

This paper propose a visual-based GUI Agent to avoid the limitations of language-based approaches. A large-scale dataset is collected through a carefully designed data collection method and used to train the visual grounding model. The paper is well-structured, clearly articulated, and demonstrates solid effectiveness in the proposed method.

**Weaknesses:**

Limitations in Completeness

1. This paper compares *UGround* with other models in Table 2 and shows UGround’s universal grounding capability. However, these methods differ in both their model settings and the training data used. An ablation study is missing to clarify the contributions of the model design (specifically the image resolution setting) and the training data to UGround's performance.

2. Same issue as in 1. This paper proposes 3 types of REs for GUI elements. An ablation study is missing to clarify the contribution of  each type of RE.

3 Related work on current GUI agents is missing from the main text.

**Questions:**

See Weaknesses.

---

> ### Author Response · Authors · 2024-11-23
> **Part 1/3: General Responses; Fair Comparison to Baseline Models**
>
> **General Response:**
>
> Thank you for the great suggestion on additional ablation studies and comparisons. We have been conducting a wide range of experiments as suggested by the reviewer, and will include them into the revised version.
>
> In this response, we provide:
>
> 1. A carefully controlled and more direct comparison with the baseline model SeeClick, controlling differences in model settings and training data.
> 2. Ablation studies on key design choices, such as image resolution and referring expression (RE) types.
> 3. Clarification of the related work discussion.
>
> For these purposes, we have trained and evaluated several model variants on ScreenSpot (Agent Setting), similar to the analyses in Section 3.5. We are also open to conducting further experiments on other benchmarks.
>
> ---
>
> > **(W1.1): Models evaluated on ScreenSpot differ in both model settings and training data.**
>
> In general, both model design and training data are essential for UGround’s good performance. Specifically, to address this concern, we introduce a new model variant, **UGround-Qwen**, fine-tuned from Qwen-VL-Chat (the same backbone used in SeeClick), using only our web-based synthetic dataset **Web-Hybrid** (processed into SeeClick's data format), to **isolate the contributions of data and model design**.
>
> | Model | Model Design | SFT data | Mobile-Text | Mobile-Icon | Desktop-Text | Desktop-Icon | Web-Text | Web-Icon | Avg |
> | ----- | ----- | ----- | ----- | ----- | ----- | ----- | ----- | ----- | ----- |
> | SeeClick | **Qwen-VL** | **Full SeeClick**: Web, Mobile, LLaVA | **81.0** | **59.8** | 69.6 | 33.6 | 43.9 | 26.2 | 52.3 |
> | UGround-Qwen | **Qwen-VL** | **Web-Hybrid** | 80.2 (-0.8) | 57.2 (-2.6) | **76.3 (+6.7)** | **39.3 (+5.7)** | **74.4 (+29.5)** | **47.1 (+20.9)** | **62.4 (+10.1)** |
> | UGround | **Ours** | **Web-Hybrid** | **89.0 (+8.8)** | **73.4 (+16.2)** | **88.1 (+11.8)** | **61.4 (+22.1)** | **84.8 (+10.4)** | **64.6 (+17.5)** | **76.9 (+14.5)** |
>
> **Training Data**:
>
> With the **same backbone** (Qwen-VL-Chat), UGround-Qwen trained on **Web-Hybrid** substantially outperforms SeeClick, with an **average absolute improvement of 10.1%**, despite SeeClick leveraging additional open-source mobile data. This experiment further confirms the quality of our synthetic data.
>
> **Model Design**:
>
> With the **same training data** (Web-Hybrid), UGround achieves a **\+14.5% absolute improvement** over UGround-Qwen, demonstrating the effectiveness of our model design.
>
> In summary, our work demonstrates strong advantages in both **model design** and **training data**. We are not comparing with **CogAgent** here because of its **lower performance** compared with SeeClick (as shown in Table 2\) despite its significantly larger model size (**18B** parameters) and data size (**140M grounding data**).
>
> Overall, these additional ablation studies further confirm the effectiveness of both our data synthesis pipeline and model design despite the simplicity of our approach which is another advantage.

---

> ### Author Response · Authors · 2024-11-23
> **Part 2/3: Ablation Studies of Model Design Decisions and Training Data**
>
> > **(W1.2):  An ablation study is missing to clarify the contributions of the model design (specifically the image resolution setting) and the training data to UGround's performance.**
>
> **Model Designs:**
>
> We focus on the model designs around image resolution here, mainly examining the following two aspects:
>
> (1) **Larger image resolution** (scaled-up large AnyRes grid settings)
>
> (2) **Dynamic resolution and aspect ratio** (compared to fixed squares)
>
> | SFT Data   | Image Res            | Mobile-Text | Mobile-Icon | Desktop-Text | Desktop-Icon | Web-Text | Web-Icon | Avg      |
> | ---------- | -------------------- | ----------- | ----------- | ------------ | ------------ | -------- | -------- | -------- |
> | Web-Hybrid | Fixed **448\*448**   | **89.4**    | 65.1        | 83.5         | 56.4         | 77.0     | 61.7     | 72.2     |
> | Web-Hybrid | Fixed **896\*896**   | 86.8        | 69.0        | 85.1         | **62.9**     | 81.4     | 57.8     | 73.8     |
> | Web-Hybrid | Fixed **1344\*1344** | 79.9        | 68.6        | 86.1         | 62.1         | 79.1     | 63.6     | 73.2     |
> | Web-Hybrid | **Dynamic (Ours)**   | 89.0        | **73.4**    | **88.1**     | 61.4         | **84.8** | **64.6** | **76.9** |
>
> #### **1\) Scaling of Image Resolution:**
>
> We scale up image resolution with fixed square sizes for convenience (**448 x 448** \-\> **896 x 896** \-\> **1344 x 1344**).
>
> As shown in the table above, larger image resolution generally improves the model performance (448 \-\> 896/1344), except for the mobile UIs, where the UIs are less dense. Notably, web and desktop UIs often contain tiny links/icons. Therefore, larger image resolution is more suitable for developing a universal model for GUIs that can handle the challenging cases.
>
> #### **2\) Dynamic Image Resolution and Aspect Ratio:**
>
> As shown in the results above, UGround benefits from dynamic image resolution supported by AnyRes, effectively adapting to varied resolutions and aspect ratios (for example, to mobile UIs or desktop UIs). This flexibility leads to improved performance across all platforms. Notably, on Desktop and Web UIs, UGround achieves comparable or better results with fewer tokens compared to the 1344 x 1344 fixed-resolution model, requiring only about 2/3 of the tokens in 16:9 scenarios.
>
> Similar findings around these two aspects are also discussed in general domains \[1, 2\], as well as some concurrent GUI works \[3, 4\]
>
> \[1\] Li, Bo, et al. LLaVA-NeXT: What Else Influences Visual Instruction Tuning Beyond Data? LLaVA Blog 2024\.
>
> \[2\] Zhang, Haotian, et al. MM1.5: Methods, Analysis & Insights from Multimodal LLM Fine-tuning. arXiv 2024
>
> \[3\] Chen, Wentong, et al. GUICourse: From General Vision Language Models to Versatile GUI Agents. arXiv 2024\.
>
> \[4\] Li, Zhangheng, et al. Ferret-UI 2: Mastering Universal User Interface Understanding Across Platforms. arXiv 2024\.
>
> **Training Data:**
>
> Regarding more fine-grained discussion on the effectiveness of our data, in addition to the new results in Part 1/3, we have also already conducted a series of experiments and analyses, as shown in Sec. 3.5 of the submitted version:
>
> 1. With our carefully curated synthesis pipeline, the web-based synthetic data **(Web-Hybrid)** can significantly boost GUI grounding performances.
>    1. Even with only **50k** screenshots and much fewer web elements, UGround outperforms the fully-trained SeeClick by over **10%,** which is trained on about 3M web and Android elements from **400K** screenshots (Figure 5).
> 2. Scaling up Web-Hybrid continuously improves the performance (Figure 5).
>    1. And from manual analysis, more data contributes to more precise locating.
>    2. It’s also noteworthy that the data is very easy to further scale up.
> 3. Web-Hybrid contributes most to the excellent performance, compared to a combination of existing datasets.  (Table 9).

---

> ### Author Response · Authors · 2024-11-23
> **Part 3/3: Ablation of RE Types; Related Work**
>
> > **(W2): Same issue as in 1\. This paper proposes 3 types of REs for GUI elements. An ablation study is missing to clarify the contribution of each type of RE.**
>
> The taxonomy for REs is an innovation in this work and was not considered in prior work (e.g., \[1-3\]).  Here we add an ablation study around Positional REs. Visual REs and functional REs are skipped because 1\) they are more or less interleaved in HTML DOMs and are not trivial to distinguish 2\) they have been largely visited by prior works.
>
> We train a new checkpoint with Web-Hybrid, where all of the positional REs are removed (but the total number of web elements remains the same). Recall that positional REs include relative positions, absolute positions, as well as contextual references, as described in Sec. 2.2.
>
> As shown in the tables below, the model trained with positional REs is generally stronger than the model trained without them.
>
> **Results from ScreenSpot (Agent Setting):**
>
> | SFT Data | Mobile-Text | Mobile-Icon | Desktop-Text | Desktop-Icon | Web-Text | Web-Icon | Avg |
> | ----- | ----- | ----- | ----- | ----- | ----- | ----- | ----- |
> | **Web-Hybrid (w/o Pos REs)** | 86.5 | 73.4 | 87.1 | 61.4 | 82.2 | **65.5** | 76.0 |
> | **Web-Hybrid** | **89.0 (+2.5)** | 73.4 | **88.1 (+1.0)** | 61.4 | **84.8 (+2.6)** | 64.6 (-0.9) | **76.9 (+0.9)** |
>
> **Results from ScreenSpot (Standard Setting):**
>
> | SFT Data | Mobile-Text | Mobile-Icon | Desktop-Text | Desktop-Icon | Web-Text | Web-Icon | Avg |
> | ----- | ----- | ----- | ----- | ----- | ----- | ----- | ----- |
> | **Web-Hybrid (w/o Pos REs)** | 72.2 | 52.0 | 72.7 | 55.0 | 76.5 | 61.2 | 64.9 |
> | **Web-Hybrid** | **75.5 (+3.3)** | **54.2 (+2.2)** | **79.9 (+7.2)** | **58.6 (+3.6)** | **77.0 (+0.5)** | **68.0 (+6.8)** | **68.8 (+3.9)** |
>
> We hypothesize that, with positional and contextual data, the model is better trained to **learn and pay attention to the context of elements**. This is generally beneficial for UI understanding, and is crucial for tasks requiring contextual understanding. In other words, training with positional RE somehow prevents models from degenerating to object detectors, which are not enough for grounding diverse referring expressions in GUIs, especially the challenging cases where visual/functional REs are not sufficient enough to help locate.
>
> \[1\] Li, Yang, et al. Widget Captioning: Generating Natural Language Description for Mobile User Interface Elements. EMNLP 2020\.
>
> \[2\] Hong, Wenyi, et al. CogAgent: A Visual Language Model for GUI Agents. CVPR 2024.
>
> \[3\] Cheng, Kanzhi, et al. SeeClick: Harnessing GUI Grounding for Advanced Visual GUI Agents. ACL 2024.
>
> > **(W3): Related work on current GUI agents is missing from the main text.**
>
> We placed the related work section in the appendix due to limited spaces. It's a fairly common practice when constrained by the spaces while the main content has comprehensively gone through related works (some prior ICLR submissions in this fashion: \[4,5\]).
>
> Specifically, we have covered 26 existing works in the main text around GUI agents, and several closely related works from general domains. For example, lines 53–55, 64–70, 83–84, 91, 171, 221–224, 348, 352, 358, and 408\. We will also further add some works in the related work section in the rebuttal version.
>
> \[4\] Jiang, Jiyan, et al. Multi-Objective Online Learning. ICLR 2023.
>
> \[5\] Yue, Xiang, et al. MAmmoTH: Building Math Generalist Models through Hybrid Instruction Tuning. ICLR 2024.

---

> > ### Comment · Reviewer_bKGa · 2024-11-25
> >
> > Thanks for your reply.
> >
> > W1:
> >
> > The model design appears to contribute more to UGround than the proposed Web-Hybrid dataset according to the table in Part 1/3. However, the proposed model design in this paper is relatively straightforward, consisting primarily of selecting an advanced base model with high input resolution.
> >
> > The paper hypothesize that visual grounding models trained only on web data may still generalize to other platforms like desktop and mobile UIs(lines 167-168). Nevertheless, UGround-Qwen underperforms compared to SeeClick on the Mobile-Text and Mobile-Icon subsets.
> >
> > W2:
> >
> > I intended to emphasize that the paper should validate the role and contribution of each component of the Web-Hybrid dataset ("The visual HTML attribute, the functional HTML attribute, and the synthesized description" mentioned in lines 201). The primary contribution of this paper is the introduction of a large scale function grounding dataset in web scenarios. However, without detailed experiments to validate the effectiveness of each component of the dataset, this paper risks appearing less innovative, as the use of LLM/MLLMs to construct dataset has already been explored in numerous previous works.
> >
> > W3:
> >
> > No further concerns.
> >
> > In summary, the core contribution of this paper (the proposed Web-Hybrid dataset) may not be sufficiently substantiated to make it a strong candidate for acceptance.

---

> > > ### Author Response · Authors · 2024-11-26
> > > **Part 1/3: W1 Follow-Ups**
> > >
> > > Firstly, we sincerely appreciate your reply regarding additional concerns. We are happy to provide further clarification.
> > >
> > > > **W1 Follow-Ups: The model design appears to contribute more to UGround than the proposed Web-Hybrid dataset according to the table in Part 1/3. However, the proposed model design in this paper is relatively straightforward, consisting primarily of selecting an advanced base model with high input resolution.**
> > > >
> > > > **The paper hypothesize that visual grounding models trained only on web data may still generalize to other platforms like desktop and mobile UIs(lines 167-168). Nevertheless, UGround-Qwen underperforms compared to SeeClick on the Mobile-Text and Mobile-Icon subsets.**
> > >
> > > It seems that there may be several misinterpretations of our results in the follow-up of W1. We are happy to clarify:
> > >
> > > | Model | Model Design | Continual SFT data | Mobile-Text | Mobile-Icon | Desktop-Text | Desktop-Icon | Web-Text | Web-Icon | Avg |
> > > | ----- | ----- | ----- | ----- | ----- | ----- | ----- | ----- | ----- | ----- |
> > > | Qwen-VL-Chat | **Qwen-VL** | None | 21.3 | 21.4 | 18.6 | 10.7 | 9.1 | 5.8 | 14.5 |
> > > | SeeClick | **Qwen-VL** | **Full SeeClick**: Web, **Mobile**, LLaVA | **81.0** | **59.8** | 69.6 | 33.6 | 43.9 | 26.2 | 52.3 |
> > > | UGround-Qwen | **Qwen-VL** | **Web-Hybrid** | 80.2 (-0.8) | 57.2 (-2.6) | **76.3 (+6.7)** | **39.3 (+5.7)** | **74.4 (+29.5)** | **47.1 (+20.9)** | **62.4 (+10.1)** |
> > > | UGround | **Ours** | **Web-Hybrid** | **89.0 (+8.8)** | **73.4 (+16.2)** | **88.1 (+11.8)** | **61.4 (+22.1)** | **84.8 (+10.4)** | **64.6 (+17.5)** | **76.9 (+14.5)** |
> > >
> > > To provide more context:
> > > 1\. We also add the grounding results of the base Qwen-VL model.
> > >
> > > 2\. We emphasize that SeeClick’s training data includes **multiple existing datasets for mobile UIs** \[1-3\], while Web-Hybrid is **entirely based on the web**.
> > >
> > > With these in mind, let’s discuss the reviewer’s concerns.
> > >
> > > 1. > The model design appears to contribute more to UGround than the proposed Web-Hybrid dataset according to the table
> > >
> > > We respectfully disagree that a 10.1% improvement (SeeClick \-\> UGround-Qwen) is much less significant than a 14.5% improvement (UGround-Qwen \-\> UGround). Both are relatively substantial. Data and modeling are orthogonal and highly complementary dimensions, and our results convincingly show that both are crucial for achieving the strong performance of UGround.
> > >
> > > 2. > The paper hypothesize that visual grounding models trained only on web data may still generalize to other platforms like desktop and mobile UIs(lines 167-168). Nevertheless, UGround-Qwen underperforms compared to SeeClick on the Mobile-Text and Mobile-Icon subsets.
> > >
> > > We also respectfully disagree with this assessment. The fact that UGround-Qwen only slightly underperforms SeeClick on mobile (while outperforming on desktop), despite SeeClick directly uses many mobile UI training datasets while UGround-Qwen’s training data is web only, **is strong evidence for the generalization of our web data**, not the other way around. This claim is also further supported by the drastic improvement of UGround-Qwen over the base model Qwen-VL-Chat on desktop and mobile (by **up to 59% absolute**).
> > >
> > > 3. > the proposed model design in this paper is relatively straightforward, consisting primarily of selecting an advanced base model with high input resolution.
> > >
> > > **Simplicity is a strength, not a weakness**. As we repeatedly emphasized throughout the paper, we present a *surprisingly simple recipe* (for both data and modeling) for training strong universal grounding models. Before our work, there was no clear evidence that training a strong universal grounding model could be so “simple”, if one considers the massive breadth and diversity of GUIs. Our work again confirms the Occam’s razor principle in a new context, that the minimalist design is usually the most generalizable.
> > >
> > > \[1\] Li, Yang, et al. Widget Captioning: Generating Natural Language Description for Mobile User Interface Elements. EMNLP 2020\.
> > >
> > > \[2\] Li, Yang, et al. Mapping Natural Language Instructions to Mobile UI Action Sequences. ACL 2020\.
> > >
> > > \[3\] Wang, Bryan, et al. Screen2Words: Automatic Mobile UI Summarization with Multimodal Learning. UIST 2021\.

---

> > > > ### Comment · Reviewer_bKGa · 2024-11-27
> > > >
> > > > 1. No further concerns.
> > > >
> > > > 2. No further concerns.
> > > >
> > > > 3. Simplicity is a strength, but not always so. I believe adopting a strong base model is simple and highly effective approach for developing a robust GUI agent, but it should not be regarded as the primary contribution. The primary contribution, in my view, comes from the developers of these strong base models, not the users.

---

> > > ### Author Response · Authors · 2024-11-26
> > > **Part 3/3: Summary**
> > >
> > > > **Summary: In summary, the core contribution of this paper (the proposed Web-Hybrid dataset) may not be sufficiently substantiated to make it a strong candidate for acceptance.**
> > >
> > > Again, we respectfully disagree with the assertion that our contribution is limited to *primarily introducing a large-scale function grounding dataset for web scenarios.*
> > >
> > > To reiterate, our contributions include:
> > >
> > > 1. We make careful arguments and a strong case for **GUI agents with human-like embodiment** that **perceive the digital world entirely visually** and **take pixel-level operations on GUIs**, and propose a **generic framework, SeeAct-V**, for building such agents by adapting from the popular SeeAct framework
> > > 2. We show that **a simple recipe**, which includes web-based synthetic data and a slight adaptation of the LLaVA architecture is surprisingly effective for GUI visual grounding.  We also train and release a universal visual grounding model, **UGround**, on the dataset.
> > > 3. We conduct the **most comprehensive evaluation for GUI agents to date**, covering six benchmarks spanning three categories: **grounding** (desktop, mobile, and web), **offline agent evaluation** (desktop, mobile, and web), and **online agent evaluation** (mobile and web). The results demonstrate:
> > >    1. **UGround** **substantially outperforms existing visual grounding models** for GUI agents across the board, by up to **20%** absolute on ScreenSpot.
> > >    2. **SeeAct-V agents with UGround** can achieve **at least comparable and often much better performance than state-of-the-art agents that use additional text-based input.**
> > >
> > > These results provide strong support for **the feasibility and promises of GUI agents that navigate the digital world as humans do.**
> > >
> > > And to the best of our knowledge, this is the first work:
> > >
> > > 1. Proposing **vision-only agents** that can work **universally in different environments**. Please also see our **response to reviewer pZoa04 (W1)** for an **in-depth discussion regarding the philosophy behind our modular design and its advantages.**
> > > 2. **Pushing the performance of vision-only GUI agents to SOTA levels** on **realistic** **live benchmarks** like AndroidWorld and Mind2Web-Live. Previously, **\[text-only or text+image observation\] and \[text-only or text+SoM grounding\] have been dominating the field** \[1-8\], with more inputs and assumptions in both observation and grounding.
> > >
> > > We hope the above response can further address the concerns and misunderstanding. If you have any questions, we are happy to discuss and provide further clarification.
> > >
> > > \[1\] Zheng, Boyuan, GPT-4V(ision) is a Generalist Web Agent, if Grounded. ICML 2024\.
> > >
> > > \[2\] He, Hongliang, WebVoyager: Building an End-to-End Web Agent with Large Multimodal Models. ACL 2024\.
> > >
> > > \[3\] Koh, Jing Yu, VisualWebArena: Evaluating Multimodal Agents on Realistic Visual Web Tasks. ACL 2024\.
> > >
> > > \[4\] Pan, Yichen, et al. WebCanvas: Benchmarking Web Agents in Online Environments. arXiv 2024\.
> > >
> > > \[5\] Kapoor, Raghav, et al. OmniACT: A Dataset and Benchmark for Enabling Multimodal Generalist Autonomous Agents for Desktop and Web. arXiv 2024\.
> > >
> > > \[6\] Xie, Tianbao, et al. OSWorld: Benchmarking Multimodal Agents for Open-Ended Tasks in Real Computer Environments. NeurIPS 2024
> > >
> > > \[7\] Li, Wei, et al. On the Effects of Data Scale on Computer Control Agents. NeurIPS 2024\.
> > >
> > > \[8\] Rawles, Christopher, et al. AndroidWorld: A dynamic benchmarking environment for autonomous agents. arXiv 2024\.

---

> > > > ### Comment · Reviewer_bKGa · 2024-11-27
> > > >
> > > > Concerns about your 3 contributions:
> > > >
> > > > 1. SeeAct-V builds upon SeeAct. SeeAct is already a GUI agent framework that can perceive the digital world and take pixel-level operations on GUIs. The main contribution of this paper is the proposal of a GUI grounding model (UGround) to assist SeeAct in grounding GUI elements.
> > > >
> > > > 2. This is the main contribution.
> > > >
> > > > 3. I respectfully disagree with the statement that "SeeAct-V agents with UGround can achieve at least comparable and often much better performance than state-of-the-art agents that use additional text-based input". Some experiments compare SeeAct-V with non-SOTA agents. For example, PaLM 2S achieves 64.8% and 80.0% on the high-level and low-level parts of the ANDROIDCONTROL benchmark, significantly outperforming SeeAct-V.[1]
> > > >
> > > > Concerns about the first work:
> > > >
> > > > 1. SeeClick is also a vision-only agent that can work universally in different environments.
> > > >
> > > > 2. I respectfully disagree. Please refer to the third point above.
> > > >
> > > > [1] Li, Wei, et al. On the Effects of Data Scale on Computer Control Agents. NeurIPS 2024.

---

> ### Author Response · Authors · 2024-11-26
> **Part 2/3: W2 Follow-Ups**
>
> > **W2 Follow-Ups: I intended to emphasize that the paper should validate the role and contribution of each component of the Web-Hybrid dataset ("The visual HTML attribute, the functional HTML attribute, and the synthesized description" mentioned in lines 201). The primary contribution of this paper is the introduction of a large scale function grounding dataset in web scenarios. However, without detailed experiments to validate the effectiveness of each component of the dataset, this paper risks appearing less innovative, as the use of LLM/MLLMs to construct dataset has already been explored in numerous previous works.**
>
> Firstly, we respectfully disagree with the assertion that our contribution is limited to *primarily introducing a large-scale function grounding dataset for web scenarios,* as just discussed above and will be elaborated more at the end of the response.
>
> We also disagree with the assertion that *the works in general domains leveraging LLM/MLLMs substantially undermine the contribution of our work*, given our careful analysis around REs in GUIs and the substantial improvement in effectiveness brought by our data compared to data in prior works in this field.
>
> Regarding the original W2:
>
> > Same issue as in 1\. This paper proposes 3 types of REs for GUI elements. An ablation study is missing to clarify the contribution of each type of RE.
>
> We have included ablation studies and discussions in the previous response.
>
> Overall, we have conducted comprehensive evaluations and ablation studies, as detailed in the paper, and provided further discussions in the last response. These substantiate **both** the **effectiveness** of **the dataset** and **its superiority to prior work**.
>
> It appears that the reviewer is suggesting an even more granular analysis of the dataset's components. While this is a valid perspective, as more and more in-depth analysis could help better understand the data, we believe it does not overshadow our existing efforts around the data, and the broader set of contributions presented in this work.

---

> > ### Comment · Reviewer_bKGa · 2024-11-27
> >
> > The ablation study is focused solely on Pos RE, excluding other types of RE. I consider the proposed Web-Hybrid dataset to be the core contribution of this paper, and conducting an in-depth analysis of it is essential.

---

> ### Author Response · Authors · 2024-11-27
> **Further Clarifications: PaLM 2S Results; SeeAct; Ablation Studies on REs; Summary**
>
> We are glad to see most concerns and misunderstandings are addressed from previous discussions. Below, we provide further clarifications.
>
> > **PaLM 2S achieves 64.8% and 80.0% on the high-level and low-level parts of the ANDROIDCONTROL benchmark, significantly outperforming SeeAct-V.**
>
> It is important to contextualize the comparison. The results referenced by the reviewer are not **apples-to-apples comparisons**. PaLM 2S was finetuned on **14K human-labeled trajectories for the AndoidControl environment**, which led to significant improvements over zero-shot PaLM 2S. A fair comparison is with the zero-shot PaLM 2S or GPT-4 settings from their paper, and our method works better. Also, **fine-tuning on agent trajectories is mostly orthogonal and complementary to the grounding aspect our work focuses on.**
>
> | Method | Model | High-level Step Acc | Low-level Step Acc |
> | :---- | :---- | ----- | ----- |
> | Zero-shot | PaLM 2S | 19.5 | 45.5 |
> | Fine-tuned | PaLM 2S | 64.8 | 80.0 |
> | Zero-shot | GPT-4 | 42.1 | 55.0 |
> | Ours (Zero-shot) | GPT-4+UGround | 46.2 | 58.0 |
>
> For Android environments, we also provided results on AndroidWorld, showcasing the SOTA performance in realistic online agent benchmarks.
>
> > **SeeAct-V builds upon SeeAct. SeeAct is already a GUI agent framework that can perceive the digital world and take pixel-level operations on GUIs. The main contribution of this paper is the proposal of a GUI grounding model (UGround) to assist SeeAct in grounding GUI elements.**
>
> There seems to be some misunderstanding about how SeeAct works.  To clarify, SeeAct  **does** **not** 1\) only take screenshots as input (it also requires text-based representations like HTML) or 2\) directly take pixel-level operations on the screen (it replies on multi-choice selection from a list of candidate elements, similar to most other existing agent designs).
>
> As introduced at lines 140-146, SeeAct-V and SeeAct have fundamental differences in both observation and grounding:
>
> |  | Grounding | Observation in Planning | Observation in Grounding | Execution |
> | :---- | :---- | :---- | :---- | :---- |
> | SeeAct-Choice | Textual Choices | Image | Image+HTML Elements | Element-Based |
> | SeeAct-Annotation | Set-of-Mark Labels | Image | Image+Marks of HTML Elements | Element-Based |
> | SeeAct-V | Pixel-Level | Image | Image | Pixel-Level |
>
> Unlike SeeAct, which depends on **HTML elements** and executes **element-based actions**, SeeAct-V perceives and operates on **raw screenshots**, leveraging **pixel-level operations** akin to human interactions.
>
> To further illustrate:
>
> **SeeAct’s Implementation**:
>
> ```python
> Elements = [...]  # a list of top-k or full HTML elements
> Index = ...       # selected by the planner
> Elements[Index].Click()
> ```
>
>
>
> **SeeAct-V’s Implementation**:
>
> ```python
> Coordinate = ...  # (x, y) from a visual grounding model
> Click(Coordinate)
> ```
>
>
>
> > **The ablation study is focused solely on Pos RE, excluding other types of RE. I consider the proposed Web-Hybrid dataset to be the core contribution of this paper, and conducting an in-depth analysis of it is essential.**
>
> While we make conceptual differentiation between visual and functional REs (which makes a useful taxonomy to illustrate the diversity and challenges), in the data synthesis process, there’s no easy way to entirely separate the two kinds. An HTML attribute (e.g., \`aria-label\`) can provide both visual and functional cues depending on the element and context. The multimodal LLM can pick up different aspects of the input when generating the RE. And even if those challenges don’t exist, doing this ablution would require us to re-generate most of our synthetic data (by removing all visual or functional information), re-train multiple models, and re-run the evaluation, which is prohibitively costly and certainly infeasible for the rebuttal period.
>
> Given that we have already addressed most of the reviewer’s concerns (both ones from the original review and new ones raised in the discussion) and this fine-grained ablation seems to be the only remaining criticism the reviewer still has, perhaps let’s do a simple thought experiment together:
>
> If we didn’t discuss this taxonomy of REs in the submission at all and everything else remains the same: **a simple data synthesis \+ modeling approach that enables human-like embodiment (visual perception \+ pixel-level operation) for GUI agents with convincing performance (across 6 benchmarks) for the first time**. How would the reviewer have assessed this work?
>
>
>
> We hope this response clarifies the remaining concerns and highlights the unique contributions of our work. Thank you again for your patient discussion.

---

> > ### Comment · Reviewer_bKGa · 2024-11-28
> >
> > 1. To support the paper's statement "SeeAct-V agents with UGround can achieve at least comparable and often much better performance than state-of-the-art agents that use additional text-based input", the authors are expected to use the best version of their agent for comparison with other SOTA agents. However, the table above shows that SeeAct-V performs better than the SOTA agents only in the zero-shot setting, where all results are too low to substantiate the authors' claim. Moreover, zero-shot GPT-4 performers much better than zero-shot PaLM 2S, suggesting that the superior performance of SeeAct-V in the zero-shot setting mainly comes from the robustness of the base model rather than the proposed UGround.
> >
> > 2. No further concerns.
> >
> > 3.
> >
> > -  "there’s no easy way to entirely separate the two kinds."
> >
> > The paper states, "We randomly select one of the following as the primary descriptor of an element: a visual HTML attribute, a functional HTML attribute, or the synthesized description by LLMs." If it is no easy to seperate them, how to randomly select one of them?
> >
> > - "If we didn’t discuss this taxonomy of REs in the submission at all and everything else remains the same..."
> >
> > If the authors did not discuss the taxonomy of REs in the submission, then nothing would change. The authors may mistakenly believe that I think this paper lacks in-depth research on the details raised. However, my concern lies in $\textbf{the degree of contribution}$. Earlier works have developed vision-based GUI agents that do not rely on GPT assistance and perform GUI navigation tasks using a single model[1], while others have constructed datasets across multiple platforms including web and mobile[2]. In contrast, this paper proposes a grounding dataset limited to the web scenario (resulting in subpar performance in mobile scenario), and develops a GUI agent based on SeeAct that still relies on GPT for planning. I actively searched for additional highlights in this paper (the design of the proposed dataset such as the taxonomy of REs) to justify acknowledging the contribution of this paper. This is why I have focused so much on the experiments.
> >
> > [1] Quanfeng Lu, et al. Gui odyssey: A Comprehensive Dataset for Cross-App Gui Navigation on Mobile Devices. arXiv 2024.
> >
> > [2] Wentong Chen, et al. GUICourse: From General Vision Language Models to Versatile GUI Agent. arXiv 2024.

---

> > > ### Author Response · Authors · 2024-12-03
> > > **Part 1/2: Comment on the Main Concern**
> > >
> > > First of all, we’d like to thank the reviewer for the patience. Reviewer engagement at ICLR this year has been generally sparser. So we appreciate the reviewer’s time and effort in getting to the bottom of things with us.
> > >
> > > It seems that the reviewer’s concern mainly comes from the positioning and contribution of this work w.r.t. the existing literature. This is quite understandable because GUI agents are such a fast-moving space with potentially confusing claims at places. Allow us to make an attempt at sorting through the development in this space (on top of what’s already discussed in the submission).
> > >
> > > Firstly, collecting training data and training a model (with visual perception only) to perform reasonably well in environments similar to the training data is not something new, nor is that the contribution we claim. Aside from papers referenced by the reviewer, Pix2Act \[1\] is great and earlier work along this line.
> > >
> > > However, it didn’t take long for the community to realize that the performance of such models often quickly degenerates outside of their training environments, i.e., under out-of-distribution settings. That’s problematic because the potential users, use cases, and environments for GUI agents are extremely broad, and we do not have the luxury to collect enough training data to make that *in-distribution*.
> > >
> > > That’s why the community got so excited about *generalist agents* powered by LLMs, a concept partially popularized by Mind2Web \[2\]. Integrating an LLM enables these agents to do reasoning (e.g., about the environmental state, retrospection on the trajectory so far, or re-planning) on the fly before having to commit to an action. They can work out of the box across environments (e.g., different websites), use cases, or even platforms (e.g., web, desktop, mobile) without in-domain training. This has quickly become the predominant paradigm for GUI agents, and that is the setting under which this work operates.
> > >
> > > In this context, when it comes to grounding, the prior wisdom is we need both visual and text-based inputs (see, e.g., SeeAct \[3\] and VisualWebArena \[4\]). That’s quite intuitive because in the generalist agent setting, where an agent needs to handle a massive number of novel situations it’s not trained for, it’s only natural to exploit whatever leverage there is. However, we believe text-based representations are not necessary, at least at inference time, and could compromise the practicality of generalist agents. A human-like embodiment should be sufficient. *This work is the first to make a careful case for human-like embodiment for generalist GUI agents and show convincing empirical support (SOTA performance across 6 benchmarks)*.
> > >
> > > Note that, concurrently or after our work, as open-weight multimodal LLMs become stronger (e.g., Qwen2-VL), there are now some attempts at fine-tuning these models with agent trajectories into generalist agents that can still generalize strongly (hopefully). That thread is still unrolling and we are enthusiastic about that, but that doesn’t overshadow the contribution of this work. We also discussed more about the philosophy behind this work in the response to reviewer pZoa.
> > >
> > > Finally, since the reviewer mentioned **GUICourse**, we add an evaluation of the Qwen-GUI model trained on GUICourse on ScreenSpot (standard setting). It performs substantially worse than both SeeClick and UGround on grounding.
> > >
> > > | Model        | Mobile-Text | Mobile-Icon | Desktop-Text | Desktop-Icon | Web-Text | Web-Icon | Avg  |
> > > | ------------ | ----------- | ----------- | ------------ | ------------ | -------- | -------- | ---- |
> > > | **Qwen-GUI** | 52.4        | 10.9        | 45.9         | 5.7          | 43.0     | 13.6     | 28.6 |
> > > | SeeClick     | 78.0        | 52.0        | 72.2         | 30.0         | 55.7     | 32.5     | 53.4 |
> > > | UGround      | 82.8        | 60.3        | 82.5         | 63.6         | 80.4     | 70.4     | 73.3 |
> > >
> > > \[1\] Shaw et al., From Pixels to UI Actions: Learning to Follow Instructions via Graphical User Interfaces. NeurIPS 2023\.
> > > \[2\] Deng et al., Mind2Web: Towards a Generalist Agent for the Web. NeurIPS 2023\.
> > > \[3\] Zheng et al., GPT-4V(ision) is a Generalist Web Agent, if Grounded. ICML 2024\.
> > > \[4\] Koh et al., VisualWebArena: Evaluating Multimodal Agents on Realistic Visual Web Tasks. ACL 2024\.

---

> ### Author Response · Authors · 2024-12-03
> **Part 2/2: Minor Clarifications about the RE Type Ablation**
>
> > **"there’s no easy way to entirely separate the two kinds." The paper states, "We randomly select one of the following as the primary descriptor of an element: a visual HTML attribute, a functional HTML attribute, or the synthesized description by LLMs." If it is no easy to seperate them, how to randomly select one of them?**
>
> We agree the original sentence could be confusing. Let’s make it more concrete. We had provided two clarifying examples in the last response:
>
> > An HTML attribute (e.g., `aria-label`) can provide both visual and functional cues depending on the element and context.
>
> > The multimodal LLM can pick up different aspects of the input when generating the RE.
>
> While we did not explicitly state in lines 204–205 (as quoted), the approach is straightforward. Although it is hard to determine whether an HTML attribute or MLLM-generated RE belongs to a specific or mixed RE types, this does not prevent us from selecting one at random. Additionally, we have included the distribution of the attributes used in Appendix F.1 (Table 6\) in the revised version. We will further revise the phrasing in the main text to avoid such misunderstandings.

---

### Official Review · Reviewer_QsLt · 2024-10-31

**Soundness:** 3
**Presentation:** 4
**Contribution:** 3
**Rating:** 8
**Confidence:** 5

**Summary:**

This paper introduces UGround, a universal GUI visual grounding model, and SeeAct-V, a vision-only framework for GUI agents. The key contributions include: (1) A novel hybrid data synthesis pipeline for creating large-scale GUI grounding training data from web sources, (2) A universal visual grounding model that achieves strong cross-platform generalization, and (3) A vision-only framework that achieves comparable or better performance than methods requiring additional textual inputs. The authors conduct comprehensive evaluations across six benchmarks spanning web, desktop, and mobile platforms, demonstrating the effectiveness of their approach in both offline and online settings.

**Strengths:**

1. Technical Innovation:
- Novel hybrid synthesis pipeline combining rule-based and LLM-based approaches
- Successful demonstration of cross-platform generalization without platform-specific training
- Effective vision-only framework that eliminates dependency on HTML/accessibility trees

2. Experimental Rigor:
- Comprehensive evaluation across multiple platforms and settings
- Strong performance improvements (up to 20% absolute improvement in standard setting)
- Thorough ablation studies on training data sources

3. Practical Impact:
- Reduces dependency on noisy and incomplete text-based representations
- Potentially more efficient due to reduced input processing requirements
- More closely mimics human interaction patterns with GUIs

4. Reproducibility:
- Clear methodology description
- Release of the largest GUI visual grounding dataset to date
- Public release of the UGround model

5. Thorough Motivation and Analysis:
- Excellent analysis of the necessity for vision-only approaches
- Clear quantification of overhead costs associated with a11y tree extraction
- Convincing demonstration of additional computational burden from processing textual information
- Well-reasoned arguments for moving away from text-based representations

**Weaknesses:**

1. Data Efficiency:
- Heavy reliance on large-scale synthetic data
- Potential redundancy in web-based training data
- Room for improvement in data deduplication and grouping

2. Limited Coverage:
- Lack of desktop UI data in training
- Incomplete handling of long-tail elements
- Platform-specific icons and elements not fully addressed

3. Dependencies:
- Reliance on external planner
- No end-to-end training with downstream tasks
- Limited standalone capability as a GUI agent

**Questions:**

Thank you for this interesting paper on vision-based web UI understanding. I have several questions and suggestions that I believe could help strengthen the work:

1. **Data Collection and Processing Details:**
   - Could you provide specific details about your webpage rendering and screenshot capture process? In particular:
     - What techniques do you use to capture content beyond the initial viewport?
     - How do you handle dynamic content that requires scrolling?
     - What is your approach for capturing interactive elements (dropdowns, expanded menus) that only appear after user interaction?
   - Documenting these technical details would help others reproduce your data collection pipeline.

2. **Desktop Application Extension:**
   - Your paper mentions potential extensions to desktop applications. Could you elaborate on:
     - What specific approaches have you considered for adapting your data collection pipeline to desktop environments?
     - How would you address the challenge of capturing UI states in desktop apps with complex interaction patterns?
     - What modifications to your current methodology would be needed to handle the diverse widget types and layouts found in desktop applications?

3. **Performance Benchmarks:**
   - Have you conducted timing experiments comparing your vision-only approach to methods that use accessibility trees, particularly for the online agent evaluation tasks?
   - Could you provide detailed end-to-end latency comparisons between your approach and traditional methods across different scenarios?
   - What are the specific performance implications of eliminating accessibility tree extraction in real-world applications?

4. **Benchmark Selection:**
   - I noticed that the evaluation doesn't include the widely-used WebArena/WorkArena/OSWorld/WindowsAgentArena benchmark. Could you explain:
     - What specific technical or methodological challenges prevented its inclusion?
     - Whether your approach is fundamentally incompatible with the these testing environments?
     - If there are plans to evaluate on this benchmark in future work?

This additional context would help readers better understand the scope and limitations of your approach, particularly regarding its applicability to different UI environments and its performance characteristics.

The current results are promising, but addressing these points would significantly strengthen the paper's contribution to the field.

---

> ### Author Response · Authors · 2024-11-24
> **Part 1/3: Data Efficiency; Limited Coverage**
>
> We would like to express our gratitude for the constructive comments and thoughtful questions. We greatly appreciate your recognition of our work's technical innovation, rigorous experimentation, practical impact, reproducibility as well as thorough analysis.
>
> We add more discussions below to help address the concerns in the weaknesses and questions.
>
> > **(W1): Data Efficiency: reliance on large-scale synthetic data; potential redundancy in web-based training data; Room for improvement in data deduplication and grouping**
>
> Our motivation centers on **leveraging web-based data synthesis** to create **scalable** and  **high-quality** grounding data. This **large-scale** synthetic data, **proven effective**, is a major contribution of our work.
>
> Regarding data efficiency, we are aware of the concern, and have taken steps to mitigate redundancy (e.g., avoiding trivial textual elements and reducing overly frequent labels, as detailed in Appendix C.1).
>
> In addition, despite the potential further improvements, the issue is **relatively minor in this study**, as the dataset is proven to be pretty effective. We have conducted studies around data in Sec. 3.5, showing that:
>
> 1. With our carefully curated synthesis pipeline, the web-based synthetic data **(Web-Hybrid)** can **significantly and much more efficiently** boost GUI grounding performances.
>    1. Even with only **50k** screenshots and much fewer web elements, UGround outperforms the fully-trained SeeClick by over **10%,** which is trained on about 3M web and Android elements from **400K** screenshots (Figure 5)—let alone CogAgent, which is trained on **140M** grounding data.
> 2. **Scaling up Web-Hybrid continuously improves the performance**.
>    1. And from manual analysis, more data contributes to more precise locating.
>    2. It’s also noteworthy that the data is very easy to further scale up.  (Figure 5).
> 3. Web-Hybrid contributes most to the excellent performance, compared to a combination of existing datasets.  (Table 9).
>
> In summary, these results have already highlighted the **efficiency and effectiveness** of our synthetic data itself, and those compared to data used in prior works. Further improvements will be considered in our future work.
>
> > **(W2): Limited Coverage: Lack of desktop UI data in training; Incomplete handling of long-tail elements; Platform-specific icons and elements not fully addressed**
>
> Our work intentionally focuses on web-based synthetic data as it is **scalable** and **demonstrates strong cross-platform generalization.**
>
> Despite the absence of desktop UI data in training, our method achieves **SOTA results** on **desktop-specific benchmarks**, with significant improvements over prior models and frameworks:
>
> **ScreenSpot (Standard Setting)**  (from Table 2\)
>
> | Model    | Desktop-Text     | Desktop-Icon     | Avg              |
> | -------- | ---------------- | ---------------- | ---------------- |
> | SeeClick | 72.2             | 30.0             | 51.1             |
> | UGround  | **82.5 (+10.3)** | **63.6 (+33.6)** | **73.1 (+22.0)** |
>
> **OmniACT** (from Table 6\)
>
> | Input         | Planner | Grounding | Action Score     |
> | ------------- | ------- | --------- | ---------------- |
> | Image \+ Text | GPT-4   | DetACT    | 17               |
> | Image         | GPT-4   | UGround   | **31.1 (+14.1)** |
>
> Based on the excellent performance of UGround’s universal GUI grounding, we hope our work (model and data) could serve as a **strong foundation** to the community, and allow future works to only focus on collecting long-tail elements like icons in specific platforms or applications.

---

> ### Author Response · Authors · 2024-11-24
> **Part 2/3: Dependencies; Data Collection and Processing Details**
>
> > **(W3): Dependencies: Reliance on external planner; No end-to-end training with downstream tasks; Limited standalone capability as a GUI agent**
>
> To be clear, UGround is developed for a **universal GUI visual grounding model**. It alone is indeed not a GUI agent. And we have largely achieved our goal and demonstrated its effectiveness in agent tasks with the SeeAct-V framework, showing SOTA or comparable results with even fewer inputs compared to end-to-end MLLMs using textual or SoM grounding on multiple benchmarks across platforms. Please also see our **response to reviewer pZoa04 (W1)** for an **in-depth discussion regarding the philosophy behind our modular designs.**
>
> While SeeAct-V integrates UGround with external MLLMs for planning,  the modular design of SeeAct-V integrating the planner and grounding model has the following advantages:
>
> 1. **Modularity**: It allows us to study and enhance UGround as a standalone grounding model, independent of specific planners. The same applies to future work on reasoning and planning models or planning frameworks.
> 2. **Flexibility**: It supports diverse MLLMs and grounding models, without specific finetuning on downstream benchmarks.
> 3. **Comparative Consistency**: By standardizing the planning stage, we can remove the confounding factor and study the impact of various grounding models and methods on the agent performance.
>
> Moreover, although our work mainly explores better grounding data and models, future work can easily adopt the data and findings in our work for training a better visually-grounded end-to-end model, as **strong grounding data has been shown to improve end-to-end models** \[1-6\].
>
> Specifically, common practices for the end-to-end training include:
>
> - Mixing grounding data in the MLLM pretraining stage before end-to-end agent task training \[4\].
> - Grounding-only training followed by training end-to-end agent data \[5,6\].
>
>
> \[1\] Cheng, Kanzhi, et al. SeeClick: Harnessing GUI grounding for advanced visual GUI agents. ACL 2024.
>
> \[2\] Liu, Junpeng, et al. Harnessing Webpage UIs for Text-Rich Visual Understanding. arXiv 2024\.
>
> \[3\] Liu, Xiao, et al. AutoGLM: Autonomous Foundation Agents for GUIs. arXiv 2024.
>
> \[4\] Hong, Wenyi, et al. CogAgent: A Visual Language Model for GUI Agents CVPR 2024\.
>
> \[5\] Chen, Wentong, et al. GUICourse: From General Vision Language Models to Versatile GUI Agents. arXiv 2024\.
>
> \[6\] Wu, Zhiyong, et al. OS-ATLAS: A Foundation Action Model for Generalist GUI Agents. arXiv 2024
>
> > **(Q1):  Data Collection and Processing Details**
>
> We appreciate the request for additional details to support reproducibility. We will include these in our revised version and will release the scripts for reproduction.
>
> All the three sub-questions can be addressed by **rendering tools** like **Playwright**. In particular:
>
> * We simulated scrolling down to capture screenshots at different heights.
> * We are aware of the interactive elements and that is a good point. They can be captured after clicking by Playwright (with rules to filter these interactive elements).
>   * For more efficient data collection, we didn't include them in Web-Hybrid. However, we do plan to incorporate them into future datasets.
>   * UGround generalizes well to the interactive elements mentioned in the question.

---

> ### Author Response · Authors · 2024-11-24
> **Part 3/3: Desktop Application Extension; Performance Benchmarks; Benchmark Selection**
>
> > **(Q2): Desktop Application Extension: adaptation of the data collection pipeline to desktop UIs**
>
> Firstly, the pipeline totally can be adapted to Desktop UIs, by adapting everything we applied to web raw data to desktop accessibility trees.
>
> We also agree one challenge is **how to efficiently capture diverse images (states)** in desktop environments with complex interaction patterns. A very straightforward, viable and cheap approach is to apply a random agent (like \[1\]), although it definitely is not optimal. A concurrent work \[2\] also applies **DFS searching** in addition to **Random Walk** for the simulation and collection.
>
> In addition, given our model’s strong overall performance and the generalization to desktop UIs, a much smaller targeted dataset focusing on unseen elements is promising to address the remaining gaps.
>
> \[1\] See an implementation of a random agent in AndroidWorld’s GitHub repo.
>
> \[2\] Wu, Zhiyong, et al. OS-ATLAS: A Foundation Action Model for Generalist GUI Agents. arXiv 2024
>
> > **(Q3):  Performance Benchmarks: timing experiments; e2e latency comparisons; performance implications of eliminating accessibility tree extraction**
>
> There are several reasons why we didn't include the exact timing and latency comparison in the paper:
>
> 1. There are multiple factors behind the latency like the network of GPT calls, what rules and implementations are applied to the processing of HTML/accessibility trees.
> 2. Although we don't need to deal with accessibility trees in the real deployment of SeeAct-V agents, the online evaluation benchmarks are largely dependent on element states for the evaluation. So we cannot fully eliminate the process in the evaluations.
>
> As for the implications of eliminating accessibility tree extraction, they have been discussed in the introduction. We are happy to explain and provide more insights here:
>
> Firstly, without accessibility trees, the input to the planner (for the observation and grounding) can be **much shorter**, for example, the input of the baseline agent in Mind2Web-Live often takes 600+ elements at a time, costing **6k+ tokens**, which is very **costly in both money and inference time**, and **potentially harms the performance** due to very long context. In comparison, a 1920\*1080 screenshot only takes **85/1105 tokens** (in low/high-resolution modes)
>
> Secondly, the elements extracted from accessibility trees are often **noisy and incomplete**. **Missing**, **redundant**, **weirdly-rendered** elements are often observed in all the environments like Mind2Web, AndroidControl, OSWorld.
>
> Additionally, besides cases where accessibility trees are unavailable, some elements are not represented by shattered elements in accessibility trees (for example, the google doc is rendered from an embedded HTML Canvas element). In those cases, only vision-only agents are capable of operating.
>
> > **(Q4):  Benchmark Selection: Whether there are specific challenges or fundamental incompatibility; Plans of evaluation in future work.**
>
> The human-like vision-only framework SeeAct-V is a **universal framework** that can be **easily adopted to every benchmark and environment**, as what we have done to the 5 agent benchmarks. There is no fundamental incompatibility. We have chosen five benchmarks based on both representativeness and task realism, spanning web, mobile, and desktop environments.
>
> However, some benchmarks may have **peculiar requirements**. Take VisualWebArena as an example. When examining the tasks and the action space in depth, some tasks require a portrait view with high resolution (such as 1280 × 2048 used in the paper), which is not perfectly compatible with the current version of UGround in terms of performance, because webpages are generally presented with a landscape orientation. We do plan to **further expand the supported resolutions** in the next version of UGround.
>
> The space of benchmarks for GUI agents has been quickly evolving, with new benchmarks coming out at a fast pace. Even though we have conducted the most comprehensive evaluation on agent benchmarks to date, inevitably we are still missing some benchmarks. We intend to continually develop our method and model and expand our evaluation to more agent benchmarks.
>
> ***
>
> We hope these responses address the reviewers’ concerns and further clarify the contributions and potential of our work. Thank you again for your thoughtful feedback.

---

> ### Author Response · Authors · 2024-11-25
> **Thanks!**
>
> Thank you very much for taking the time to review our additional experiments and discussions. We sincerely appreciate your thoughtful feedback and are truly grateful for your recognition of our efforts.

---

### Official Review · Reviewer_pZoa · 2024-11-04

**Soundness:** 3
**Presentation:** 3
**Contribution:** 3
**Rating:** 8
**Confidence:** 4

**Summary:**

This paper introduces a vision-only approach to GUI agents that mimics how humans interact with interfaces through visual perception and pixel-level operations. Moving away from conventional reliance on HTML and accessibility trees, the authors develop SeeAct-V, a framework for human-like GUI interaction, and UGround, a visual grounding model trained on the largest GUI grounding dataset to date (19M elements). Through comprehensive evaluation across six benchmarks, they demonstrate that UGround outperforms existing grounding models by up to 20%, while SeeAct-V agents match or exceed state-of-the-art agents that use additional text-based input.

**Strengths:**

- This paper contributes to reframes GUI interaction as a pure visual grounding problem, challenging the conventional wisdom that additional textual representations are necessary.
- The authors develop a novel way to generate diverse referring expressions (REs) by categorizing them into visual, positional, and functional types. And they introduce an innovative hybrid data synthesis pipeline that combines rule-based and LLM-based approaches
- This paper includes a comprehensive agent evaluation covering Three platforms (web, desktop, mobile), Three evaluation settings, (grounding, offline agent, online agent) six different benchmarks.

**Weaknesses:**

- UGround relies on an external LLM planner and cannot operate independently as a GUI agent without training on downstream tasks. When combined with the Scaling Curve in Figure 5 on Web-Hybrid, it becomes challenging to enhance agent performance by merely increasing grounding data. Instead, improvements depend on the external LLM planner, which may limit the potential of the SeeAct-V framework.

- In the current model architecture, the authors have increased the input image size to 36 grids of CLIP@224. This results in a large number of image tokens ((224/14)^2 * 36 = 9,216), leading to inefficiency. The authors could conduct an ablation study to determine if this resolution is necessary, considering the cost of image tokens.

**Questions:**

- What factors contribute to the higher Completion Rate (CR) of the image-based model (GPT-4o with UGround) compared to the text-based model on Mind2Web-Live (Table 7), even though the Success Rate (SR) is comparable or lower? Could the grounding method (UGround vs. Choice) impact CR and SR differently across input modalities?

---

> ### Author Response · Authors · 2024-11-24
> **Part 1/2: Dependency on External LLM Planner; Scaling Limitation**
>
> > **(W1) Dependency on External LLM Planner; Scaling Limitation**
>
> Thank you for bringing up this very interesting question. This gives us a good chance to share more about our philosophy behind the modular design of SeeAct-V and UGround.
>
> When it comes to agent designs, the current wisdom, by and large, is to train a *monolithic LLM* (e.g., CogAgent, SeeClick, many ongoing supervised fine-tuning efforts for enhancing ‘agentic behaviors’, and perhaps Anthropic’s recent computer use agent as well). At a philosophical level, part of the goal of SeeAct-V is to challenge that status quo and advocate a *modular design* for language agents instead.
>
> A fundamental challenge of language agents comes from the complex, dynamic, and often idiosyncratic environments where they operate. Just take web agents as an example. There are over a billion websites out there. Each website could have an infinite number of states and be constantly changing (driven by updates in the backend databases). There is also a considerable amount of highly idiosyncratic semantics in each environment, e.g., uncommon icons, jargon, and counter-intuitive designs.
>
> As a result, although we are still at the early stage of agent research, we hold a strong belief that a monolithic model, no matter how large and strong it will become, is unlikely to capture all the complexity and idiosyncrasy of all the environments. In order to develop a generalist agent that can generalize reliably across environments, we need to adopt a *modular system design* and synergistically orchestrate a foundation model (e.g., GPT-4o) with multiple specialized modules that serve different functionalities.
>
> Grounding is one of such capabilities for which a specialized module makes a lot of sense. Fundamentally, grounding is about understanding domain-specific semantics and creating a map between that and natural language (that LLMs know about). A dedicated module makes it much easier to capture idiosyncratic semantics and easily adapt to different domains (e.g., imagine fine-tuning the grounding model instead of the foundation model for a domain). The grounding model can then supply domain-specific semantics to the foundation model and lead to a \`1+1\>2\` kind of system efficacy. That’s a fundamental motivation for the design of SeeAct-V and this whole work.
>
> Our design also offers several practical advantages:
>
> 1. **Modularity**: It allows us to study and enhance UGround as a standalone grounding model, independent of specific planners.
> 2. **Flexibility**: It supports diverse MLLMs and grounding models, without specific finetuning on downstream benchmarks.
> 3. **Comparative Consistency**: By standardizing the planning stage, we can remove the confounding factor and study the impact of various grounding models and methods on the agent performance.
>
> As demonstrated in our experiments, SeeAct-V with UGround does surpass the performance of end-to-end MLLMs (whether they are using textual or SoM grounding). In addition, we want to note that training end-to-end approaches requires a large amount of quality data on agent trajectories (essentially combining planning and grounding) for downstream tasks, which is very challenging to obtain in general.

---

> ### Author Response · Authors · 2024-11-24
> **Part 2/2: Image Resolution; Results on Mind2Web-Live**
>
> > **(W2) the authors have increased the input image size to 36 grids of CLIP@224. This results in a large number of image tokens ((224/14)^2 \* 36 \= 9,216), leading to inefficiency. The authors could conduct an ablation study to determine if this resolution is necessary, considering the cost of image tokens.**
>
> We appreciate the insightful suggestion to investigate the efficiency w.r.t. image size. In our response to **Reviewer bKGa**, we have included **comprehensive ablation studies on image resolution and dynamic resolution.** Please check the details there, and we will include them in the revised version.
>
> Key findings include:
>
> 1. **Higher Resolutions Enhance Dense UIs**: Higher image resolutions generally improve grounding performance, especially for dense or complex UIs with small elements.
> 2. **Universal Model**: Higher resolutions are essential for handling challenges like tiny links or icons, contributing to the goal of developing a universal model that can work on diverse GUIs.
> 3. **Dynamic Resolutions**: Supporting dynamic resolutions improves flexibility, reduces token cost, and enhances performance.
>
> We would also like to point out that our resolution choices are consistent with practices and findings in recent works (\[1-6\]). That said, we agree with the reviewer that how to improve the efficiency of processing a large number of image tokens is an important problem and should be investigated in the future. We will include these findings and discussions in our revised version.
>
> \[1\] Hong, Wenyi, et al. CogAgent: A Visual Language Model for GUI Agents. CVPR 2024\.
>
> \[2\] Zhang, Haotian, et al. MM1. 5: Methods, analysis & insights from multimodal LLM fine-tuning. arXiv 2024\.
>
> \[3\] Chen, Wentong, et al. GUICourse: From General Vision Language Models to Versatile GUI Agents. arXiv 2024\.
>
> \[4\] Li, Zhangheng, et al. Ferret-UI 2: Mastering Universal User Interface Understanding Across Platforms. arXiv 2024\.
>
> \[5\] Anthropic. Introducing computer use, a new Claude 3.5 Sonnet, and Claude 3.5 Haiku. Anthropic Blog 2024\.
>
> \[6\] Wu, Zhiyong, et al. OS-ATLAS: A Foundation Action Model for Generalist GUI Agents. arXiv 2024
>
> > **(Q1) What factors contribute to the higher Completion Rate (CR) of the image-based model (GPT-4o with UGround) compared to the text-based model on Mind2Web-Live (Table 7), even though the Success Rate (SR) is comparable or lower? Could the grounding method (UGround vs. Choice) impact CR and SR differently across input modalities?**
>
> To clarify, **Completion Rate (CR)** measures **partial task success** (the portion of completed key nodes), while **Success Rate (SR)** requires **full task completion**. So the discrepancy between CR and SR happens (e.g., also shown in the official results of GPT-4o and Gemini-1.5 Pro in \[1\]). We had also conducted several rounds of testing and got consistent results.
>
> A simple example (0/1: a failure or success of an evaluation key node, assuming each task contains two key nodes here):
> **Agent 1**: (0,0), (1,0,), (1,0), (1,0) \-\> **CR: 0.375** SR: 0
>
> **Agent 2**: (1,1),  (0,0), (0,0), (0,0) \-\> CR: 0.25 **SR: 0.25**
>
> Overall, we find that with UGround, **grounding is less of a bottleneck now**, and **planning has become a more outstanding bottleneck** (see, e.g., Figure 4 in the submission). The tasks in Mind2Web-Live, which are extracted from Mind2Web, tend to require stronger planning and a longer planning horizon than tasks from most other agent benchmarks. As a result, stronger grounding doesn’t necessarily linearly correlate with a higher end-to-end success rate. Oftentimes, the planner would make a mistake at a certain step in a long trajectory and fail the whole task, despite the stronger grounding model successfully completing more key nodes (so higher completion rates).
>
> \[1\] Pan, Yichen, et al. WebCanvas: Benchmarking Web Agents in Online Environments. arXiv 2024\.

---

> > ### Comment · Reviewer_pZoa · 2024-12-02
> > **Thanks for your detailed response.**
> >
> > Thank you for your insightful response. I would like to maintain my positive rating.

---

> ### Author Response · Authors · 2024-12-02
> **Thanks!**
>
> Thank you very much for taking the time to review our additional experiments and discussions. We are deeply grateful for your recognition of our work!

---

### Meta-Review · Area_Chair_EEZs · 2024-12-20

**Metareview:**

This reviewers generally liked the work, which presents a vision-only approach for GUI agents, using a visual grounding model (UGround) trained on a large synthetic dataset (Web-Hybrid). The authors argue this approach is more human-like and avoids the noise and overhead of text-based representations. The motivation makes sense, and the shift from text (DOM) based visual grounding to vision only approach has began a few years ago, e.g., Spotlight paper by Li & Li from ICLR. The strengths of the work are strong empirical results that show significant performance improvement and comprehensive evaluation as well as detailed analysis. On the other hand, the reviewers criticized the work on dependency over an external planner and data efficiency and limited coverage with the synthetic dataset.

Overall:
This paper presents a compelling case for vision-only GUI agents, supported by strong empirical results and comprehensive evaluation. The authors effectively address the concerns raised by the reviewers, providing further analysis and clarification. The work advanced our understanding of developing more human-like and efficient GUI agents.

**Additional Comments On Reviewer Discussion:**

The reviewers raised a number of questions, and the authors addressed these questions well, e.g., the rationale of a modular design, the issues with data synthesis, the generalization of the findings. The reviewers also wanted further clarification on details such as image resolution and referring expressions. Overall, the reviewers are mostly happy with the author rebuttal. Reviewer bKGa still has reservations, although most of the reviewer's concerns have been addressed in the rebuttal.

---

### Decision · Program_Chairs · 2025-01-22

Accept (Oral)